

# Scenario set-up and the new CMIP6-based climate-related forcings provided within the third round of the Inter-Sectoral Model Intercomparison Project (ISIMIP3b, group I and II)

Katja Frieler[1,2], Stefan Lange[1], Jacob Schewe[1], Matthias Mengel[1], Simon Treu[1,2], Christian Otto[1], Jan Volkholz[1], Christopher P.O. Reyer[1], Stefanie Heinicke[1], Colin Jones[3], Julia L. Blanchard[4], Cheryl S. Harrison[5], Colleen M. Petrik[6], Tyler D. Eddy[7], Kelly Ortega-Cisneros[8], Camilla Novaglio[4], Ryan Heneghan[9], Derek P. Tittensor[10], Olivier Maury[11], Matthias Büchner[1], Thomas Vogt[1], Dánnell Quesada Chacón[1], Kerry Emanuel[12], Chia-Ying Lee[13], Suzana J. Camargo[13,14], Jonas Jägermeyr[14,15,1], Sam Rabin[16,a,b], Jochen Klar[1], Iliusi D. Vega del Valle[1], Lisa Novak[1], Inga J. Sauer[1], Gitta Lasslop[17], Sarah Chadburn[18], Eleanor Burke[19], Angela Gallego-Sala[20], Noah Smith[21], Jinfeng Chang[22], Stijn Hantson[23], Chantelle Burton[19], Anne Gädeke[1], Fang Li[24], Simon N Gosling[25], Hannes Müller Schmied[17,26], Fred Hattermann[1], Thomas Hickler[17], Rafael Marcé[27,], Don Pierson[28], Wim Thiery[29], Daniel Mercado-Bettín[27], Robert Ladwig[30], Ana I. Ayala[28], Matthew Forrest[17], Michel Bechtold[31], Robert Reinecke[32], Inge de Graaf[33], Jed O. Kaplan[34], Alexander Koch[35], Matthieu Lengaigne[11]

Affiliations:

[1]Potsdam Institute for Climate Impact Research, 14473 Potsdam, Germany

[2]University of Potsdam, Institute for Environmental Science and Geography, 14476 Potsdam, Germany

[3]National Centre for Atmospheric Science and School of Earth and Environment, University of Leeds, Leeds, LS29JT, UK

[4]Institute for Marine and Antarctic Studies, University of Tasmania, Hobart, Tasmania, Australia

[5]Department of Ocean and Coastal Science and Center for Computation and Technology, Louisiana State University, Baton Rouge, Louisiana, USA

[6]Scripps Institution of Oceanography, University of California San Diego, CA, USA

[7]Centre for Fisheries Ecosystems Research, Fisheries & Marine Institute, Memorial University, St. John's, NL, Canada

[8]Marine and Antarctic Research for Innovation and Sustainability, Department of Biological Sciences, University of Cape Town, Rondebosch, Cape Town, 7701, South Africa

[9]School of Environment and Science, Griffith University, Brisbane, Queensland, Australia

[10]Department of Biology, Dalhousie University, Halifax, Nova Scotia, Canada, B3H 4R2

[11]IRD, Univ Montpellier, CNRS, Ifremer, INRAE, MARBEC, Sète, France

[12]Lorenz Center, Massachusetts Institute of Technology, Cambridge, MA, USA

[13]Lamont-Doherty Earth Observatory, Columbia University, Palisades, New York, USA

[14]Columbia Climate School, Columbia University, New York, NY 10025, USA

[15]NASA Goddard Institute for Space Studies, New York, NY 10025, USA

[16]Climate and Global Dynamics Laboratory, National Center for Atmospheric Research Boulder, CO 80302, USA

[17]Senckenberg Leibniz Biodiversity and Climate Research Centre (SBiK-F), Frankfurt am Main, Germany.

[18]Department of Mathematics, University of Exeter, Exeter UK

[19]Met Office Hadley Centre, Fitzroy Road, Exeter, UK

[20]Geography Department, University of Exeter, Exeter, UK



[21]College of Engineering, Mathematics and Physical Sciences, University of Exeter, Exeter EX4 4QF, UK.
[22]College of Environmental and Resource Sciences, Zhejiang University, Hangzhou, China
[23]Faculty of Natural Sciences, Universidad del Rosario, Bogotá, Colombia
[24]International Center for Climate and Environment Sciences, Institute of Atmospheric Physics, Chinese
Academy of Sciences, Beijing, China
[25]School of Geography, University of Nottingham, Nottingham, UK
[26]Institute of Physical Geography, Goethe University Frankfurt, Frankfurt am Main, Germany
[27]Blanes Centre for Advanced Studies (CEAB-CSIC), Blanes, Spain
[28]Department of Ecology and Genetics, Uppsala University, Norbyvägen 18 D, 752 36 Uppsala, Sweden
[29]Vrije Universiteit Brussel, Department of Water and Climate, Brussels, Belgium
[30]Department of Ecoscience, Aarhus University, C.F. Møllers Allé 3, 8000 Aarhus C, Denmark
[31]KU Leuven, Department of Earth and Environmental Sciences, Leuven, Belgium
[32]Johannes Gutenberg-University Mainz, Mainz, Germany
[33]Earth Systems and Global Change Group, Wageningen University and Research, Wageningen, The
Netherlands
[34]Department of Earth Sciences and Institute for Climate and Carbon Neutrality, The University of Hong
Kong, Hong Kong
[35]Simon Fraser University, Burnaby, British Columbia, CA
[a]formerly at: Institute of Meteorology and Climate Research / Atmospheric Environmental Research,
Karlsruhe Institute of Technology, Garmisch-Partenkirchen, Germany
[b]formerly at: Department of Environmental Sciences, Rutgers University, New Brunswick, New Jersey,
USA
*Correspondence to:* Katja Frieler (katja.frieler@pik-potsdam.de)
**Abstract.** This paper describes the climate-related forcings (CRFs) provided within the 'b' part of the
third simulation round of the Inter-Sectoral Impact Model Intercomparison Project (ISIMIP3b). While
ISIMIP3a comprises historical impact models simulations forced by observational CRF and direct human
forcings (DHF), the ISIMIP3b CRFs are based on climate model simulations generated within the sixth
phase of the Coupled Model Intercomparison Project (CMIP6). In a first set of experiments (ISIMIP3b,
group I) the CMIP6-based CRFs for the historical period are combined with historical observation-based
DHF also considered in ISIMIP3a (e.g. land use patterns, water and agricultural management, and fishing
efforts). These group I simulations allow for the quantification of impacts of historical climate change by
comparison to simulations where the observational DHF are combined with simulated pre-industrial
CRFs. In addition, the impacts of observed changes in CRFs can be compared to the impacts of simulated
changes in CRFs by comparing the ISIMIP3a simulations to the ISIMIP3b, group I simulations. The second
group of experiments (ISIMIP3b, group II) comprises future projections assuming constant observational
direct human forcings at 2015 levels to estimate the impact of climate change given today's direct
human influences for the low emission scenario SSP1-2.6, the high and the very high emission scenarios
SSP3-7.0, SSP5-8.5, respectively. The very high emissions scenarios and the assumption of fixed present
day direct human forcings particularly allow for testing the scalability of impacts in terms of global
temperature change. The provided CRFs comprise atmospheric $CO_2$ and $CH_4$ concentrations,



atmospheric and oceanic climate data, coastal water levels, tropical cyclone tracks and their associated wind speed and precipitation fields. In addition to the CRFs data, this paper describes the experiments belonging to group I and II and the rationale behind them. Another set of future projections accounting for changing DHFs (ISIMIP3b, group III) is in preparation and will be described in another paper.

**Introduction**

This is the second paper of a series of three papers describing the experiments of the third simulation round of the Inter-Sectoral Impact Model Intercomparison Project ISIMIP (isimip.org). The project provides a common scenario framework for cross-sectorally consistent climate impact simulations. In its third round it covers i) model evaluation and climate impact attribution experiments based on observation-based climate and direct human forcings (ISIMIP3a, first paper, (Frieler et al., 2023)), ii) climate impact simulations driven by simulated climate-related forcings based the sixth phase of the Coupled Climate Model Intercomparison Project (CMIP6) assuming ISIMIP3a observational DHF in the historical period and fixed 2015 DHF for the future simulations (ISIMIP3, group I+II, this paper), and iii) an upcoming set of CMIP6-based future projections where DHF vary according to given Shared Socioeconomic Pathways (SSPs) (no adaptation scenarios) and in response to climate change impacts (adaptation scenarios) (ISIMIP3b, group III). So while this paper only describes the ISIMIP3b climate-related forcings, the third paper will only address the DHFs that are still under development while the CRF of the group III simulations will be identical to the future CRF described here.

Similar to the Coupled Model Intercomparison Project (CMIP) (Eyring et al., 2016) all simulations will be freely available (https://data.isimip.org/) to allow for follow-up analysis. The consistent design of the simulations does not only allow for the comparison of climate impact simulations within each sector, but also enables the bottom-up integration of impacts across sectors. Thus, it provides a unique basis for the estimation of the effects of climate change on, e.g., the economy, displacement and migration, health, or water quality resolving the mechanisms along different impact channels and fully exploiting the process-understanding represented in the biophysical impact models.

Compared to ISIMIP2b, the ISIMIP3b CRF represent the following updates: i) climate forcing data based on phase 6 of the Coupled Model Intercomparison Project (CMIP6) (Eyring et al., 2016) and post-processed by an improved bias adjustment and statistical downscaling method (see section **3.2**), and ii) large ensembles of potential realisations of tropical cyclone tracks, wind and precipitation fields derived from two different modelling approaches assuming CMIP6 boundary conditions, while in ISIMIP2b only one approach was used and precipitation fields were not included. In addition, we plan to provide coastal water levels at high temporal resolution (upcoming). The approach to generate the data is also described here.

The development of the ISIMIP3b protocol was coordinated by the ISIMIP-Cross-Sectoral Science Team (CSST) at the Potsdam Institute for Climate Impact Research (PIK) along the same decision process as for ISIMIP3a (Frieler, submitted 2023).



This paper is accompanied by a simulation protocol (*ISIMIP3b Simulation Protocol*, 2023) providing all technical details such as file and variable naming conventions, as well as sector-specific output variables to be reported by the participating modelling teams. This paper refers to the protocol version of December 21st, 2023. However, as the protocol may still be updated due to addition of new variables, correction of errors, or the inclusion of new sectors, contributors to ISIMIP should always refer to protocol.isimip.org for the most up to date reference for planned impact model simulations.

The ISMIP3a and ISIMIP3b protocols have been jointly developed and participation in ISIMIP3 requires contribution to both ISIMIP3a and ISIMIP3b, using the same impact model versions in order to allow for the evaluation of the impact models future projections in ISIMIP3b.

In the following, we describe the rationale behind the individual scenario set-ups (section **1**). We then introduce the individual climate-related forcing data sets in the second section covering atmospheric climate data including lighting and tropical cyclones tracks, wind and precipitation fields; ocean data; coastal water levels; and atmospheric $CO_2$ as well as $CH_4$ concentrations.

## 1 Experiments and underlying rationale

The selection of ISIMIP3b scenarios (see **Figure 1**) was generally motivated by the aim to i) capture a wide range of possible futures from low to high emission scenarios, ii) the availability of climate model simulations, and iii) the provision of a long baseline simulation assuming pre-industrial climate conditions that allows for a robust estimation of reference return levels of extreme events. Given recent mitigation efforts, some estimates of recoverable coal reserves, and decreasing prices for renewable energies the emissions underlying SSP5-8.5 have been criticised for not representing a meaningful 'business as usual scenario' (Hausfather & Peters, 2020). Therefore, within ISIMIP SSP5-8.5 is not considered a 'business as usual scenario', but rather a worst case scenario. Furthermore, its strong warming signal allows testing to what degree the simulated impacts of climate may scale with global mean temperature, which could potentially lead to translating impacts to other emission scenarios. In addition, even under lower emission scenarios, global warming levels as the ones reached under SSP5-8.5 in 2100 will only be reached later in time as long as emissions are not reduced to zero. These impacts of high warming levels would not be captured when only considering lower emission scenarios ending in 2100. In response to the discussion, the 'average no climate protection policy' SSP3-7.0 (Hausfather & Peters, 2020) has been added to the ISIMIP3b protocol. However, SSP3-7.0 has not been designed as a business as usual scenario, either. Instead it is based on rather extreme assumptions about land use changes and aerosol emissions e.g. leading to a scaling of precipitation with global mean temperature that diverges from the scaling identified in the other scenarios (Shiogama et al., 2023).

All ISIMIP experiments are determined by the underlying set of CRFs and DHF, where each package of CRF and DHF has a specific label that will then be included in the output file names to allow for an identification of the experiments they belong to. The individual experiments are defined by the combination of both types of forcing data sets, where the associated specifiers are indicated in brackets in the subheadings naming the experiments (CRF specifier + DHF specifier). The different combinations



of the default sets of ISIMIP3b CRFs ('picontrol', 'historical', 'ssp126', 'ssp370', and 'ssp585') and DHF
('histsoc', '2015soc', '1901soc', '1850soc', 'nat', and '2015soc-from-histsoc') are sketched in **Figure 1** and
described in more detail below.

The CRF forcing data described in this paper are mandatory; i.e. if impact models consider this forcing,
the specified dataset must be used; if an alternative input data set is used instead, the run cannot be
considered an ISIMIP3, group I + II simulation. The DHF for the historical period is identical to the
ISIMIP3a DHF listed in **Table 1** of (Frieler, submitted 2023) where we also indicate whether the data set
is mandatory or optional. Optional forcing data could be used but is not mandatory. In addition, the
protocol includes a set of sensitivity experiments that are described as deviations from the default runs
and labelled by the baseline CRF and DHF settings and a third specifier indicating the deviation from this
default setting. The ISIMIP3b group I+II sensitivity runs include experiments with fixed levels of
atmospheric $CO_2$ concentrations ('2015co2'), high levels in $CO_2$ concentrations in combination with low
levels of climate change ('ssp585co2'), and runs with lightning data that vary in response to climate
change ('varlightning'), while lightning is fixed at present day levels in the default runs. These sensitivity
experiment runs are not depicted in Figure 1 but listed in **Table 2**.


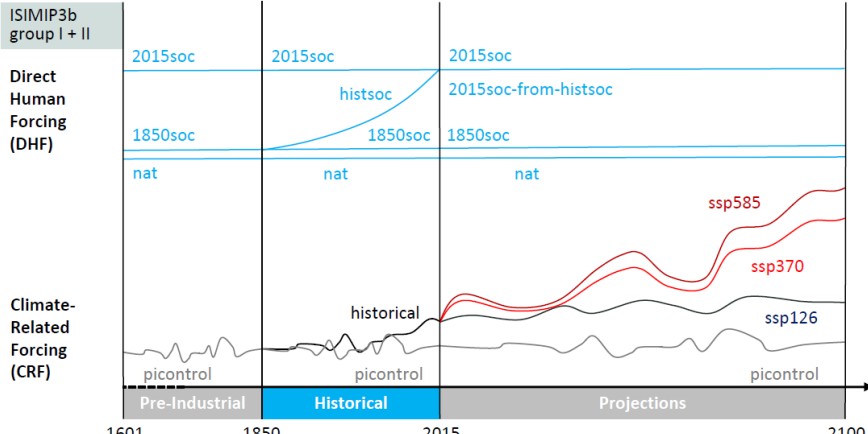


**Figure 1:** Illustration of the default ISIMIP3b forcing data sets. Each ISIMIP3b experiment is defined by a
combination of a CRF data set with a DHF data set. The considered combinations are listed in **Table 2**
and the underlying rationale is described in section **1.1** and **1.2**. **Table 1** lists all data sets defining the
considered CRFs while the DHFs are based on the same datasets as in ISIMIP3a. Potentially required
spin-up procedures are not included in the Figure, but described in section **1.1**.

The ISMIP3b simulations are divided into two groups. Group I comprises the simulations from 1601 -
1849 (pre-industrial) and 1850 - 2014 (historical) assuming simulated pre-industrial and historical CRFs
and different constant ('nat', '1850soc', and '2015soc') or varying ('histsoc') levels of DHF based on the



same observational data used in ISIMIP3a (see **Figure 1**). Group II comprises the future projections
assuming constant 2015 levels of DHF (see **Figure 1**) including a baseline with pre-industrial CRF (grey
line in the future projections part of **Figure 1**). All experiments are introduced in more detail below
(section **1.1** for group I and **1.2** for group II).
In contrast to ISIMIP3a, the CRFs provided for ISIMIP3b currently only comprise atmospheric (see
section **2.1**) and oceanic climate data (see section **2.4**), tropical cyclone tracks with associated wind and
precipitation fields (see section **2.2**), and $CO_2$ and methane concentration (see section **2.5**). We do not
yet provide associated coastal water levels (see section **2.2.3** for planned work), and lightning data (see
**Table 5**). Impact simulations that rely on the missing forcings cannot be generated within ISIMIP3b yet,
but we are currently developing their setup and will provide the forcings as soon as possible. The
ISIMIP3b atmospheric and oceanic climate data is derived from five different General Circulation Models
generated within the Coupled Model Intercomparison project, phase 6 (CMIP6).
**Table 1: Climate-Related Forcing datasets for ISIMIP3b.**

| Forcing | Status | Source, description |
|---|---|---|
| Climate-related forcings ('picontrol', 'historical', 'ssp126', 'ssp370', 'ssp585') | | |
| Atmospheric forcings ('picontrol', 'historical', 'ssp585', 'ssp370', 'ssp126') | | |
| Gridded atmospheric climate forcing | mandatory | Bias-adjusted data (pre-industrial climate, historical climate, and future projections for the SSP1-2.6, SSP3-7.0, and SSP5-8.5 scenarios) generated by GFDL-ESM4, IPSL-CM6A-LR, MPI-ESM1-2-HR, MRI-ESM2-0, and UKESM1-0-LL within CMIP6, see section **2.1** |
| Local atmospheric climate forcing for lakes | mandatory | Atmospheric data extracted from the data sets above for 72 lakes that have been identified within the lake sector as locations (grid cell of the ISIMIP 0.5° grid, ISIMIP3 local lake sites) where models can be calibrated based on observed temperature profiles and hypsometry within ISIMIP3b (depth and area). |
| Tropical cyclone tracks with wind and precipitation fields | mandatory | Available on request (see section **2.2**), samples of synthetic tropical cyclone tracks derived from the five CMIP6 GCMs considered within ISIMIP generated by two different statistical downscaling approaches, see section **2.2**.<br><br>MIT approach (Emanuel et al., 2008):<br>historical climate from IPSL-CM6A-LR, MPI-ESM1-2-HR, UKESM1-0-LL and GFDL-ESM4 (all 1850-2014), and from MRI-ESM2-0 (1950-2014). Future climate: ssp370 and ssp585 (2015-2100) from IPSL-CM6A-LR, MPI-ESM1-2-HR, MRI-ESM2-0, UKESM1-0-LL, and ssp585 (2061-2100) from GFDL-ESM4. |



| | | Two different configurations (SD and CRH, see section **2.2**) of the Columbia HAZard model (CHAZ, (Lee et al., 2018)): pre-industrial climate (1601-2100) from GFDL-ESM4, IPSL-CM6A-LR, MPI-ESM1-2-HR, MRI-ESM2-0, and UKESM1-0-LL. historical climate (1850-2014) from GFDL-ESM4, IPSL-CM6A-LR, MPI-ESM1-2-HR, MRI-ESM2-0, and UKESM1-0-LL future climate (2015-2100): ssp126, ssp370, ssp585 from GFDL-ESM4, IPSL-CM6A-LR, MPI-ESM1-2-HR, MRI-ESM2-0, and UKESM1-0-LL |
|---|---|---|
| Lightning | mandatory | Flash Rate Monthly Climatology from (Cecil, 2006), not changing with climate change |
| **Oceanic forcings ('picontrol', 'historical', 'ssp585', 'ssp370', 'ssp126')** | | |
| Oceanic climate forcing | mandatory | Uncorrected data (pre-industrial climate, historical climate, and future projections for the SSP1-2.6, SSP3-7.0, and SSP5-8.5 scenarios) generated by GFDL-ESM4, IPSL-CM6A-LR, MPI-ESM1-2-HR, and UKESM1-0-LL within CMIP6, see section **2.4** |
| **Coastal water levels** | | |
| Coastal water levels | mandatory | Not available yet, but we plan to provide hourly water levels derived from the atmospheric forcings described above combined with long-term sea-level trends; see section **2.3.** |
| **Atmospheric composition or fluxes** | | |
| Atmospheric $CO_2$ concentration | mandatory | (Büchner & Reyer, 2022) based on the following sources: 1850-2005: (Meinshausen et al., 2011); 2006-2014: Global annual CO2 from NOAA Global Monthly Mean $CO_2$ ((Lan et al., 2023); 2015-2100: (Meinshausen et al., 2020) |
| Atmospheric $CH_4$ concentration | mandatory | (Büchner & Reyer, 2022) based on the following sources: 1850-2014: (Meinshausen et al., 2017); 2015-2100: (Meinshausen et al., 2020) |
| **Climate-Related Forcings for the sensitivity experiment 'varlightning', using above forcing data except for:** | | |
| **Lightning data ('varlightning')** | | |
| Varying lightning according to climate change | mandatory | Lightning data has been generated for the ssp126, ssp370, and ssp850 climate projections from UKESM1-0-LL (Kaplan et al., 2023) |
| **Climate-Related Forcings for the 'de-biased' sensitivity experiment** | | |
| **Global oceanic forcings** | | |



| Oceanic forcings based on de-biased atmospheric forcings | mandatory | Not available yet, simulated by the ocean biogeochemistry model ocean-biogeochemistry NEMO-PISCES forced by a de-biased version of the IPSL-CM6A-LR-based atmospheric forcing (see section **2.4.2**) |
|---|---|---|
| Regional oceanic forcings | | |
| De-biased oceanic forcing based on observed oceanic data for individual variables and regions | mandatory | Not centrally provided, see section **2.4.3** |



## 1.1 ISIMIP3b, group I: Climate-model based impact model simulations for the period from 1601 to 2015


The group I experiments cover the years 1601-1849 with pre-industrial CRFs ('picontrol') and fixed 1850
direct human forcings ('1850soc') described in the grey column 3 of the ISIMIP3b scenario **Table 2** as
well as the subsequent years 1850-2014 considering pre-industrial and historical climate-related forcings
('picontrol' or 'historical') and different assumptions about direct human forcings ('histsoc', '2015soc',
'1850soc', and 'nat') as described in the grey column 4 of **Table 2**. The reasoning behind the individual
experiments are introduced below.

**Pre-industrial reference simulations (picontrol + histsoc, picontrol + 2015soc, picontrol +**
**2015soc-from-histsoc, picontrol + 1850soc, picontrol + nat; default):** To estimate the impacts of
historical and future changes in the CRFs, the protocol includes reference simulations based on
pre-industrial CRFs and DHF identical to those considered in the climate change scenario runs. In order
to allow for the fitting of extreme value distributions, e.g. to estimate reference 100 year return levels of
certain impacts, the runs are designed to  includes the generation of large samples (at least 250 years)
of impact distributions distribution based on stable pre-industrial CRFs (picontrol) and constant DHFs
(see 'picontrol + 1850soc', 'picontrol + 2015soc', and 'picontrol + nat' experiments in  **Table 2**).
In addition, the protocol includes a reference experiment for the historical period (1850-2014) with DHF
changing over time (histsoc) and  1850-2014 pre-industrial CRF (picontrol), while fixed 2015 DHF is
considered afterwards (2015-2100) ('picontrol + 2015soc-from-histsoc'). This run may be different from
the 'picontrol + 2015soc' simulation for this time window because of the lagged effects of increasing
DHF from 1850 to 2014. The 'histsoc' DHF is identical to ISIMIP3a  (Frieler, submitted 2023).
The complete pre-industrial reference runs are divided in two parts. Only the first parts from the start
until 2014 belong to group I (grey fields in the table), while the second parts covering the period
2015-2100 belong to group II (red parts of the table).

Comparing these reference simulations to the scenario experiments using historical CRFs (historical +
histsoc, historical + 2015soc, historical + 1850soc, historical + nat; default (see below)) allows for the



estimation of the effects of simulated historical climate change conditional on the assumed DHF. The historical climate-related forcing ('historical') starts from the pre-industrial climate simulation in 1850, i.e. the 'picontrol' and 'historical' versions of the experiments have a common starting point. As some impact indicators may have 'internal' trends not necessarily forced by external drivers (e.g. re-growth of forests), the comparison of the 1850-2014 impact simulations forced by the 'historical' CRF to parallel simulations using the 'picontrol' CRF is more appropriate to estimate the effects of historical climate change than comparing an early period of the historical impact simulation to the end of the historical period.

For models requiring a spin-up, the 'picontrol' CRFs should be used in combinations with DHF i) at 1850 levels to spin-up for the '1850soc' and 'histsoc' experiments, ii) at 2015 levels to initialise the '2015soc' experiment, and iii) set to zero to start the 'nat' experiments. For the spin-up the 'picontrol' CRF should be copied as often as needed. The 'picontrol + 1850soc' run from 1601-1849 is part of the regular experiments that should be reported and hence the spin-up has to be finished before this pre-industrial period.

To allow for a quantification of the impacts of the anthropogenic CRFs, we also support historical reference simulation assuming only natural CRF ('hist-nat' simulations generated within the Detection and Attribution Model Intercomparison Project (DAMIP) as sub-MIP of CMIP6, (Gillett et al., 2016) by providing the associated bias-adjusted CRF as secondary climate input data (Lange et al., 2023). However, associated simulations are not an official part of ISIMIP3b and not described in the associated protocol.

**Standard historical simulations based on historical climate-related forcing and observed changes in direct human forcing (historical + histsoc; default):** The historical simulations (1850-2014) are forced by historical ('historical') CRFs and DHFs evolving according to observations (ISIMIP3a 'histsoc' DHF). The ISIMIP3b 'historical + histsoc' experiment is comparable to the default 'obsclim + histsoc' run used in ISIMIP3a but based on simulated CRFs. The simulated climate is different from the observed realisation due to differences in the internal variability of the observed and simulated historical climate and potential deficits in the climate model simulations. A comparison between the default ISIMIP3b 'historical + histsoc' impact model simulations to the associated ISIMIP3a results allows for a quantification of the effects of the discrepancies between the observed and simulated CRFs on the considered impact indicators. This experiment can be initialised from the spin-up of the associated pre-industrial reference simulation in case a spin-up is needed.

**Simulations with historical climate-related forcing and fixed 2015 direct human forcing (historical + 2015soc; default):** This historical experiment is similar to the standard historical experiment except that it is forced by fixed 2015 DHF. It is introduced into the 'first priority' scenario-set-up to generate an ensemble of historical cross-sectorally consistent impact simulations that is as large as possible by not excluding impact models that are not able to handle varying DHF. If a spin-up is needed the experiment can be initialised from the spin-up of the associated pre-industrial reference simulation (picontrol + 2015soc, default) described at the beginning of this section.



**Simulations with historical climate-related forcing and fixed 1850 direct human forcing (historical + 1850soc; default):** This historical experiment is also similar to the standard historical experiment but it is forced by the fixed 1850 DHFs. It corresponds to the 'obsclim + 1901soc' simulation of ISIMIP3a. Here in ISIMIP3b we consider the year 1850 instead of 1901 used in ISIMIP3a as this is the year where the 'historical' climate simulations with observed natural and human forcings start, i.e. a branch from the pre-industrial climate simulations assuming constant pre-industrial forcings ('picontrol'). If a spin-up is required it does not have to be newly generated as it is identical to the spin-up for the default 'picontrol + 1850soc', 'picontrol + histsoc', and 'historical + histsoc' experiments and described in the beginning of this section.

**Simulations with historical climate-related forcing and no direct human forcings (historical + nat; default):** Considering no direct human forcings (nature run) allows quantifying the effect of the simulated historical climate change conditional on otherwise natural conditions, i.e. no direct human influences on land use, water management etc.. This experiment is introduced as a companion experiment to the 'obsclim + nat' simulations of ISIMIP3a. The comparison with the three historical simulations with constant direct human forcings allows testing, to what degree the impact of climate change on the simulated natural or human systems is conditional on the underlying direct human forcing. This experiment is only included for the biomes and fisheries and marine ecosystems fisheries sectors as models from other sectors usually need some basic information such as vegetation patterns that are not available for natural-only conditions. The biomes models generate their own natural-only vegetation patterns based on their dynamic representation of vegetation. A spin-up does not have to be newly generated but is identical to the spin-up for the 'picontrol + nat' experiment described above.

**'De-biased' sensitivity simulations within the marine ecosystems and fisheries sector (FishMIP) with de-biased historical oceanic forcings and no or histsoc direct human forcings (historical + nat, historical + histsoc; de-biased):** So far, the default oceanic forcing is not bias-adjusted as globally the observational data are to sparse to be used in a similar empirical way as for the bias-adjustment of the atmospheric forcing. However, the biases in the forcing are expected to also induce biases in the historical and future impact simulations. To quantify these effects and to test a suggested bias-adjustment method based on comprehensive ocean-biogeochemistry model simulations forced by bias-adjusted atmospheric forcings we include a sensitivity experiment where the default climate-related forcing is replaced by input data generated by a dynamical de-biasing approach (Lengaigne et al. 2025) using the NEMO-PISCES physical-biogeochemical ocean model (Madec, 2015), which is the oceanic component of the IPSL-CM6A-LR climate model. Thus, the forcing data will first be generated for IPSL-CM6A-LR, but later extended to other ISIMIP-GCMs as described in subsection **2.4.2.** In contrast, the oceanic forcing for the regional component of the marine ecosystems and fisheries sector have been bias-adjusted by regional observational oceanic data as described in subsection **2.4.3.** In this case most models only use the bias-adjusted inputs and not the raw ones. Nevertheless the experiments are labeled as 'de-biased' sensitivity experiments to ensure a consistent naming across scales.





## 1.2 ISIMIP3b, group II: Climate-model based future impact model simulations with constant 2015 direct human forcings

The ISIMIP3b, group II simulations comprise a set of future impact projections (2015-2100) using fixed levels of direct human forcings as considered in the historical simulations ('2015soc', '1850soc', and 'nat') or reached at the end (2014) of the historical period in the 'historical + histsoc' runs ('2015soc-from-histsoc'). These runs are described in the red cells of **Table 4**.

**Pre-industrial reference simulations (picontrol + 2015soc, picontrol + 2015soc-from-histsoc, picontrol + 1850soc, picontrol + nat; default):** These simulations are included into the ISIMIP3, group II part of the protocol to allow for the estimation of the effect of climate change by comparing the future impact projections to simulations assuming the same background DHF but pre-industrial levels of CRF (see description of baseline simulations in section **1.1**).

**Future impact projections assuming SSP-RCP-based climate-related forcings starting from 'historical + histsoc' simulations (ssp126 + 2015soc-from-histsoc, ssp370 + 2015soc-from-histsoc, ssp585 + 2015soc-from-histsoc; default):** These runs are a continuation of the group I 'historical + histsoc' simulations assuming fixed 2015 direct human forcings for the future. Note that this experiment is different from the experiment with fixed 2015 DHF for the future starting from the 'historical + 2015soc' group I experiment (see description below).

These experiments have been introduced to describe the impacts of different scenarios of changes in the climate-related systems on today's natural systems and societies, i.e. assuming present day population levels and distributions, land use patterns, water, and agricultural management measures etc.. In many cases, the projected changes in natural and human systems can be interpreted as the pure effect of the prescribed changes in the climate-related systems. However, they could also partly result from lagged effects of the historical changes in DHFs ('histsoc'), CRF ('historical'), or natural temporal trends induced e.g. by re-growth of forests. To be able to separate natural trends from the effects of changing CRFs, these simulations can be compared to reference impact simulations with pre-industrial climate-related forcings forced with the same direct human forcings ('picontrol + 2015soc-from-histsoc', see description in group I section).

**Future impact projections assuming SSP-RCP-based climate-related forcings starting from historical simulations with constant 2015 direct human forcings (ssp126 + 2015soc, ssp370 + 2015soc, ssp585 + 2015soc; default):** These experiments continue the 'historical + 2015soc' experiments from ISIMIP3b, group I using direct human forcings held constant at 2015 levels for the historical period. Although the DHF in the future period is identical to the future simulations described above, the difference in historical forcing may affect the impact simulations in the future period. These simulations are also considered first priority as some of the impact models may not be able to handle varying direct human forcings and therefore can only perform these experiments. Models participating in the





'2015soc-from-histsoc' experiments described above are also asked to complete the '2015soc' runs to
generate an as large as possible ensemble of consistent impact model simulations.

**Future impact projections assuming SSP-RCP-based climate-related forcings starting from historical**
**simulations assuming constant 1850 direct human forcings (ssp126 + 1850soc, ssp370 + 1850soc,**
**ssp585 + 1850soc; default):** These experiments continue the default 'historical + 1850soc' experiments
considered in ISIMIP3b, group I. They are included to estimate the impact of changes in the
climate-related systems conditional on 1850 levels of direct human forcings that can be compared to the
impact conditional on today's levels of direct human forcings ('2015soc').

**Future impact projections assuming SSP-RCP-based climate-related forcings starting from historical**
**simulations assuming no direct human forcings (ssp126 + nat, ssp370 + nat, ssp585 + nat; default):**
These experiments continue the default 'historical + nat' experiments in ISIMIP3b, group I. They are
included to estimate the effect of changes in the climate-related systems (here climate change itself and
increasing $CO_2$ concentrations) assuming no direct human forcings.

**$CO_2$ sensitivity simulations (ssp126 + 2015soc-from-histsoc, ssp370 + 2015soc-from-histsoc, ssp585 +**
**2015soc-from-histsoc, ssp585 + 2015soc, ssp585 + 1850soc, ssp585 + nat; 2015co2):** To separate the
effects of increasing atmospheric $CO_2$ concentrations from the effects of other changes in the
climate-related systems, the ISIMIP3b protocol includes sensitivity experiments where atmospheric $CO_2$
concentrations are held constant at 2015 levels. For SSP1-2.6 and SSP3-7.0, they are only introduced as
deviations from the default '2015soc-from-histsoc' experiments while for SSP5-8.5 the effect can also be
quantified conditional on all levels of direct human influences considered in the previous experiments.
**Future lightning sensitivity simulations (ssp126 + 2015soc-from-histsoc, ssp370 +**
**2015soc-from-histsoc, ssp585 + 2015soc-from-histsoc; varlightning):** To study the effects of future
changes in lightning flash rates as opposed to using a stationary lightning climatology, the ISIMIP3b
protocol includes sensitivity experiments where future lightning flash rates change along the RCPs. The
future lightning data sensitivity experiment is introduced as a deviation from the default
'2015soc-from-histsoc" experiment and only for one climate model (UK-ESM). This sensitivity
experiment has been introduced for the fire sector.
**Climate sensitivity simulations under high levels of $CO_2$ (ssp126 + 2015soc-from-histsoc, ssp585co2):**
To study the effects of high atmospheric $CO_2$ concentration without accompanying changes in climate,
the ISIMIP3b protocol includes a sensitivity experiment where the atmospheric $CO_2$ concentration are
prescribed according to RCP8.5, while the other climate-related forcings, in particular the atmospheric
forcings are from SSP1-2.6. The future climate sensitivity experiment is introduced as a deviation from
the default 'ssp126 + 2015soc-from-histsoc' experiment. This sensitivity experiment has been
introduced for the peat sector.
**'De-biased' sensitivity simulations within the marine ecosystems and fisheries sector (FishMIP) with**
**de-biased oceanic forcings and no or 2015soc direct human forcings for reference simulations based**
**on pre-industrial oceanic forcing (picontrol + nat, picontrol + 2015soc-from-histsoc; de-biased) and the**





**associated simulations accounting for different levels of climate change (ssp126 + nat, ssp370 + nat,**
**ssp858 + nat, ssp126 + 2015soc-from-histsoc, ssp370 + 2015soc-from-histsoc, ssp585 +**
**2015soc-from-histsoc):** These simulations represent the future extensions of the 'de-biased' group I
simulations described above. They are designed to test the dynamical bias-adjustment suggested for the
global oceanic forcings under different levels of climate change (ssp126, ssp370, ssp585). The regional
impact projections within the sector are also based on de-biased oceanic forcings and are therefore also
labeled as 'de-biased' sensitivity experiments to ensure a consistent labeling across scales.


| Experiment | Short description | Period: Pre-industrial 1601-1849 | Period: Historical 1850-2014 | Period: Future 2015-2100 |
|---|---|---|---|---|
| **pre-industrial control** 2015soc-from-histsoc **1st priority** | **CRF:** No changes in the climate-related systems, $CO_2$ and $CH_4$ fixed at 1850 levels | **picontrol** | **picontrol** | **picontrol** |
| | **DHF:** Varying management before 2015, then fixed at 2015 levels thereafter | **1850soc** | **histsoc** | **2015soc-from-histsoc** |
| **pre-industrial control** 2015soc **1st priority** | **CRF:** No changes in the climate-related systems, $CO_2$ and $CH_4$ fixed at 1850 levels | Does not have to be simulated as the following periods already provide 251 simulation years assuming stable baseline CRF and DHF. ensi | **picontrol** | **picontrol** |
| | **DHF:** Fixed at 2015 levels for all periods | | **2015soc** | **2015soc** |
| **pre-industrial control** 1850soc **2nd priority** | **CRF:** No changes in the climate-related systems, $CO_2$ and $CH_4$ fixed at 1850 levels | Does not have to be simulated as the following periods already provide 251 simulation years assuming stable baseline CRF and DHF. | **picontrol** | **picontrol** |
| | **DHF:** Fixed at 1850 levels for all periods | | **1850soc** | **1850soc** |



| | | | | |
|---|---|---|---|---|
| **pre-industrial control**<br><br>nat<br><br>**2nd priority** | **CRF:** No changes in the climate-related systems, CO₂ and CH₄ fixed at 1850 levels | Does not have to be simulated as the following periods already provide 251 simulation years assuming stable baseline CRF and DHF. | **picontrol** | **picontrol** |
| | **DHF:** No direct human influences | | **nat** | **nat** |
| **RCP2.6**<br><br>2015soc-from-hist soc<br><br>**1st priority** | **CRF:** Simulated historical changes in climate-related systems, CO₂ and CH₄ concentrations as observed in the historical period, then simulated SSP1-2.6 changes in the climate-related systems | Identical to picontrol + 1850soc run described above | **historical** | **ssp126** |
| | **DHF:** Varying management before 2015, then fixed at 2015 levels thereafter | | **histsoc** | **2015soc-from-histsoc** |
| **RCP2.6**<br><br>2015soc<br><br>**1st priority** | **CRF:** Simulated historical changes in climate-related systems, CO₂ and CH₄ concentrations as observed in the historical period, then simulated SSP1-2.6 changes in the climate-related systems | Identical to "picontrol + 2015soc" run | **historical** | **ssp126** |
| | **DHF:** Fixed at 2015 levels for all periods | | **2015soc** | **2015soc** |
| **RCP2.6**<br><br>1850soc<br><br>**2nd priority** | **CRF:** Simulated historical changes in climate-related systems, CO₂ and CH₄ concentrations as observed in the | Identical to "picontrol + 1850soc" run | **historical** | **ssp126** |



| | | | | |
|---|---|---|---|---|
| | historical period, then simulated SSP1-2.6 changes in the climate-related systems | | | |
| | **DHF:** Fixed at 1850 levels for all periods | | **1850soc** | **1850soc** |
| **RCP2.6**<br><br>nat<br><br>**2nd priority** | **CRF:** Simulated historical changes in climate-related systems, CO$_2$ and CH$_4$ concentrations as observed in the historical period, then simulated SSP1-2.6 changes in the climate-related systems | Identical to "picontrol + nat" run | **historical** | **ssp126** |
| | **DHF:** No direct human influences | | **nat** | **nat** |
| **CO$_2$ sensitivity RCP2.6**<br><br>2015soc-from-hist soc<br><br>**2nd priority** | **CRF:** Simulated historical changes in climate-related systems, CO$_2$ and CH$_4$ concentrations as observed in the historical period, then simulated SSP1-2.6 changes in the climate-related systems but fixed 2015 CO$_2$ concentrations | Identical to "picontrol + 1850soc" run | "histsoc" version of the historical period of the RCP2.6 experiment, as described above | **ssp126**<br><br>**Sensitivity experiment: 2015co2** |
| | **DHF:** Varying management before 2015, then fixed at 2015 levels thereafter | | | **2015soc-from-histsoc** |
| **RCP7.0** | **CRF:** Simulated historical changes in climate-related systems, CO$_2$ and CH$_4$ | Identical to "picontrol + 1850soc" run | "histsoc" version of the historical period of the | **ssp370** |



| | | | | |
|---|---|---|---|---|
| 2015soc-from-hist soc<br><br>**1st priority** | concentrations as observed in the historical period, then simulated SSP3-7.0 changes in the climate-related systems | | RCP2.6 experiment | |
| | **DHF:** Varying management before 2015, then fixed at 2015 levels thereafter | | | **2015soc-from-histsoc** |
| **RCP7.0**<br><br>2015soc<br><br>**1st priority** | **CRF:** Simulated historical changes in climate-related systems, $CO_2$ and $CH_4$ concentrations as observed in the historical period, then simulated SSP3-7.0 changes in the climate-related systems | Identical to "picontrol + 2015soc" run | Identical to "historical + 2015soc" run described above | **ssp370** |
| | **DHF:** Fixed at 2015 levels for all periods | | | **2015soc** |
| **RCP7.0**<br><br>1850soc<br><br>**2nd priority** | **CRF:** Simulated historical changes in climate-related systems, $CO_2$ and $CH_4$ concentrations as observed in the historical period, then simulated SSP3-7.0 changes in the climate-related systems | Identical to "picontrol + 1850soc" run | Identical to "historical + 1850soc" run described above | **ssp370** |
| | **DHF:** Fixed at 1850 levels for all periods | | | **1850soc** |



| | | | | |
|---|---|---|---|---|
| **RCP7.0**<br><br>nat<br><br>**2nd priority** | **CRF:** Simulated historical changes in climate-related systems, $CO_2$ and $CH_4$ concentrations as observed in the historical period, then simulated SSP3-7.0 changes in the climate-related systems | Identical to "picontrol + nat" run | Identical to "historical + nat" run described above | **ssp370** |
| | **DHF:** No direct human influences | | | **nat** |
| **CO$_2$ sensitivity RCP7**<br><br>2015soc-from-hist soc<br><br>**2nd priority** | **CRF:** Simulated historical changes in climate-related systems, $CO_2$ and $CH_4$ concentrations as observed in the historical period, then simulated SSP3-7.0 changes in the climate-related systems but $CO_2$ concentrations fixed at 2015 levels | Identical to "picontrol + 1850soc" run | Identical to "historical + histsoc" run described above | **ssp370**<br><br>**Sensitivity experiment: 2015co2** |
| | **DHF:** Varying management before 2015, then fixed at 2015 levels thereafter | | | **2015soc-from-histsoc** |
| **RCP8.5**<br><br>2015soc-from-hist soc<br><br>**1st priority** | **CRF:** Simulated historical changes in climate-related systems, $CO_2$ and $CH_4$ concentrations as observed in the historical period, then simulated SSP5-8.5 changes in | Identical to "picontrol + 1850soc" run | Identical to "historical + histsoc" run described above | **ssp585** |



| | the climate-related systems | | | |
| --- | --- | --- | --- | --- |
| | **DHF:** Varying management before 2015, then fixed at 2015 levels thereafter | | | **2015soc-from-histsoc** |
| **RCP8.5** 2015soc **1st priority** | **CRF:** Simulated historical changes in climate-related systems, $CO_2$ and $CH_4$ concentrations as observed in the historical period, then simulated SSP5-8.5 changes in the climate-related systems | Identical to "picontrol + 2015soc" run | Identical to "historical + 2015soc" run described above | **ssp585** |
| | **DHF:** Fixed at 2015 levels for all periods | | | **2015soc** |
| **RCP8.5** 1850soc **2nd priority** | **CRF:** Simulated historical changes in climate-related systems, $CO_2$ and $CH_4$ concentrations as observed in the historical period, then simulated SSP5-8.5 changes in the climate-related systems | Identical to "picontrol + 1850soc" run | Identical to "historical + 1850soc" run described above | **ssp585** |
| | **DHF:** Fixed at 1850 levels for all periods | | | **1850soc** |
| **RCP8.5** nat **2nd priority** | **CRF:** Simulated historical changes in climate-related systems, $CO_2$ and $CH_4$ concentrations as observed in the historical period, | Identical to "picontrol + nat" run | Identical to "historical + nat" run | **ssp585** |





| | | | | |
|---|---|---|---|---|
| | then simulated SSP5-8.5 changes in the climate-related systems | | | |
| | **DHF:** No direct human influences | | | **nat** |
| **CO₂ sensitivity RCP8.5**<br><br>2015soc-from-hist soc<br><br>**1st priority** | **CRF:** Simulated historical changes in climate-related systems, CO₂ and CH₄ concentrations as observed in the historical period, then simulated SSP5-8.5 changes in the climate-related systems but CO₂ concentrations fixed at 2015 levels | Identical to "picontrol + 1850soc" run | Identical to "historical + histsoc" run | **ssp585**<br><br>**Sensitivity experiment: 2015co2** |
| | **DHF:** Varying management before 2015, then fixed at 2015 levels thereafter | | | **2015soc-from-histsoc** |
| **CO₂ sensitivity RCP8.5**<br><br>2015soc<br><br>**1st priority** | **CRF:** Simulated historical changes in climate-related systems, CO₂ and CH₄ concentrations as observed in the historical period, then simulated SSP5-8.5 changes in the climate-related systems, but CO₂ concentrations fixed at 2015 levels | Identical to "picontrol + 2015soc" run | Identical to "historical + 2015soc" run | **ssp585**<br><br>**Sensitivity experiment: 2015co2** |
| | **DHF:** Fixed at 2015 levels for all periods | | | **2015soc** |



| | | | | |
|---|---|---|---|---|
| **CO₂ sensitivity RCP8.5**<br><br>1850soc<br><br>**2nd priority** | **CRF:** Simulated historical changes in climate-related systems, CO₂ and CH₄ concentrations as observed in the historical period, then simulated SSP5-8.5 changes in the climate-related systems, but CO₂ concentrations fixed at 2015 levels | Identical to "picontrol + 1850soc" run | Identical to "historical + 1850soc" run | **ssp585**<br><br>**Sensitivity experiment: 2015co2** |
| | **DHF:** Fixed at 1850 levels for all periods | | | **1850soc** |
| **CO₂ sensitivity RCP8.5**<br><br>nat<br><br>**1st priority** | **CRF:** Simulated historical changes in climate-related systems, CO₂ and CH₄ concentrations as observed in the historical period, then simulated SSP5-8.5 changes in the climate-related systems, but CO₂ concentrations fixed at 2015 levels | Identical to "picontrol + nat" run | Identical to "historical + nat" run | **ssp585**<br><br>**Sensitivity experiment: 2015co2** |
| | **DHF:** No direct human influences | | | **nat** |
| **Lightning sensitivity RCP2.6**<br><br>2015soc-from-histsoc<br><br>**2nd priority** | **CRF:** Simulated historical changes in climate-related systems, CO₂ and CH₄ concentrations as observed in the historical period, then simulated SSP1-2.6 changes in the climate-related systems including | Identical to "picontrol + 1850soc" run | Identical to "historical + histsoc" run | **ssp126**<br><br>**Sensitivity experiment: varlightning** |



| | | | | |
|---|---|---|---|---|
| | future lightning which in the default case is considered fixed at climatological levels | | | |
| | **DHF:** Varying management before 2015, then fixed at 2015 levels thereafter | | | **2015soc-from-histsoc** |
| **Lightning sensitivity RCP7.0** 2015soc-from-hist soc **2nd priority** | **CRF:** Simulated historical changes in climate-related systems, $CO_2$ and $CH_4$ concentrations as observed in the historical period, then simulated SSP3-7.0 changes in the climate-related systems including future lightning which in the default case is considered fixed at climatological levels | Identical to "picontrol + 1850soc" run | Identical to "historical + histsoc" run | **ssp370** **Sensitivity experiment: varlightning** |
| | **DHF:** Varying management before 2015, then fixed at 2015 levels thereafter | | | **2015soc-from-histsoc** |
| **Lightning sensitivity RCP8.5** 2015soc-from-hist soc **2nd priority** | **CRF:** Simulated historical changes in climate-related systems, $CO_2$ and $CH_4$ concentrations as observed in the historical period, then simulated SSP5-8.5 changes in the climate-related systems including future lightning which in the default | Identical to "picontrol + 1850soc" run | Identical to "historical + histsoc" run | **ssp585** **Sensitivity experiment: varlightning** |



| | | | | |
|---|---|---|---|---|
| | case is considered fixed at climatological levels | | | |
| | **DHF:** Varying management before 2015, then fixed at 2015 levels thereafter | | | **2015soc-from-histsoc** |
| **Climate sensitivity, RCP2.6 with RCP8.5 CO₂** <br><br> 2015soc-from-histsoc <br><br> **2nd priority** | **CRF:** Simulated historical changes in climate-related systems, CO₂ and CH₄ concentrations as observed in the historical period, then CO₂ evolves according to SSP5-8.5 while all other CRFs change according to default SSP1-2.6 forcing data | Identical to "picontrol + 1850soc" run | Identical to "historical + histsoc" run | **ssp126** <br><br> **Sensitivity experiment: ssp585co2** |
| | **DHF:** Varying management before 2015, then fixed at 2015 levels thereafter | | | **2015soc-from-histsoc** |
| **Bias sensitivity, de-biased oceanic data for pre-industrial control** <br><br> nat <br><br> **2nd priority** | **CRF:** De-biased pre-industrial oceanic forcing, CO₂ fixed at 1850 levels | Not covered | **picontrol** | **picontrol** <br><br> **Sensitivity experiment: de-biased** |
| | **DHF:** no direct human influences | Not covered | **nat** | **nat** |
| **Bias sensitivity, de-biased oceanic data for SSP1-2.6** <br><br> nat <br><br> **2nd priority** | **CRF:** De-biased simulated historical oceanic forcing, then de-biased simulated SSP1-2.6 oceanic forcing | Not covered | **historical** | **ssp126** <br><br> **Sensitivity experiment: de-biased** |
| | **DHF:** no direct human influences | Not covered | **nat** | **nat** |





| | | | | |
|---|---|---|---|---|
| **Bias sensitivity, de-biased oceanic data for SSP3-7.0**<br><br>nat<br><br>**2nd priority** | **CRF:** De-biased simulated historical oceanic forcing, then de-biased simulated SSP3-7.0 oceanic forcing | Not covered | **historical** | **ssp370**<br><br>**Sensitivity experiment: de-biased** |
| | **DHF:** no direct human influences | Not covered | **nat** | **nat** |
| **Bias sensitivity, de-biased oceanic data for SSP5-8.5**<br><br>nat<br><br>**2nd priority** | **CRF:** De-biased simulated historical oceanic forcing, then de-biased simulated SSP5-8.5 oceanic forcing | Not covered | **historical** | **ssp585**<br><br>**Sensitivity experiment: de-biased** |
| | **DHF:** No direct human influences | Not covered | **nat** | **nat** |
| **Bias sensitivity, de-biased oceanic data for pre-industrial control**<br><br>2015soc-from-histsoc<br><br>**2nd priority** | **CRF:** De-biased pre-industrial oceanic forcing, $CO_2$ fixed at 1850 levels | Not covered | **picontrol** | **picontrol**<br><br>**Sensitivity experiment: de-biased** |
| | **DHF:** Varying direct human influences before 2015, then fixed at 2015 levels thereafter | Not covered | **histsoc** | **2015soc-from-histsoc** |
| **Bias sensitivity, de-biased oceanic data for SSP1-2.6**<br><br>2015soc-from-histsoc<br><br>**2nd priority** | **CRF:** De-biased simulated historical oceanic forcing, then de-biased simulated SSP1-2.6 oceanic forcing | Not covered | **historical** | **ssp126**<br><br>**Sensitivity experiment: de-biased** |
| | **DHF:** Varying direct human influences before 2015, then fixed at 2015 levels thereafter | Not covered | **histsoc** | **2015soc-from-histsoc** |



| Bias sensitivity, de-biased oceanic data for SSP3-7.0<br><br>2015soc-from-histsoc<br><br>**2nd priority** | **CRF:** De-biased simulated historical oceanic forcing, then de-biased simulated SSP3-7.0 oceanic forcing | Not covered | **historical** | **ssp370**<br><br>**Sensitivity experiment: de-biased** |
| | **DHF:** Varying direct human influences before 2015, then fixed at 2015 levels thereafter | Not covered | **histsoc** | **2015soc-from-histsoc** |
| Bias sensitivity, de-biased oceanic data for SSP5-8.5<br><br>2015soc-from-histsoc<br><br>**2nd priority** | **CRF:** De-biased simulated historical oceanic forcing, then de-biased simulated SSP5-8.5 oceanic forcing | Not covered | historical | **ssp585**<br><br>**Sensitivity experiment: de-biased** |
| | **DHF:** Varying direct human influences before 2015, then fixed at 2015 levels thereafter | Not covered | **histsoc** | **2015soc-from-histsoc** |

**Table 2: ISIMIP3b climate-model based experiments.** The table provides a comprehensive list of all ISIMIP3b, group I (grey) and group II (red) experiments defined by the assumed climate-related forcings (CRF) and direct human forcings (DHF). Here the climate-related forcings are only described by the climate (oceanic and atmospheric) and $CO_2$ forcings as we do not provide coastal water levels yet.

## 2 Climate-related forcing data

### 2.1 Bias-adjusted and statistically downscaled atmospheric climate forcing

For ISIMIP3b we provide the daily atmospheric forcings for the same variables as in ISIMIP3a on the default 0.5° grid (see **Table 3**). These variables are from the output of CMIP6 climate model simulations, selected and processed as described below. We use the climate simulations from the picontrol (for pre-industrial conditions), historical (for historical conditions), ssp126, ssp370, and ssp585 (for future conditions under the scenarios SSP1-2.6, SSP3-7.0, and SSP5-8.5, respectively) CMIP6 experiments.



**Table 3:** Climate-related atmospheric forcing data provided within ISIMIP3b. The upper limits of pr and prsn correspond to 600 mm day-1 and 300 mm day-1, respectively, while the lower and upper limits of tas, tasmax and tasmin correspond to −90°C and +70°C, respectively.

| Variable | Variable specifier | Unit (maximum range, inner bounds if considered) | Resolution | Datasets |
|---|---|---|---|---|
| Near-Surface Relative Humidity | **hurs** | % ([1, 100], [0.01, 99.99]) | 0.5° grid, daily | Bias-adjusted and downscaled from GFDL-ESM4, IPSL-CM6A-LR, MPI-ESM1-2-HR, MRI-ESM2-0, and UKESM1-0-LL simulations generated for CMIP6. |
| Near-Surface Specific Humidity | **huss** | kg kg-1 ([0.0000001, 0.1]) | 0.5° grid, daily | Derived from bias-adjusted and downscaled hurs, ps, and tas from GFDL-ESM4, IPSL-CM6A-LR, MPI-ESM1-2-HR, MRI-ESM2-0, and UKESM1-0-LL simulations generated for CMIP6. |
| Precipitation (including snowfall) | **pr** | kg m-2 s-1 ([0, 600/86400], [0.1/86400, ∞[) | 0.5° grid, daily | Bias-adjusted and downscaled from GFDL-ESM4, IPSL-CM6A-LR, MPI-ESM1-2-HR, MRI-ESM2-0, and UKESM1-0-LL simulations generated for CMIP6. |
| Snowfall | **prsn** | kg m-2 s-1 ([0, 300/86400])<br><br>Maximum range and inner bounds of unitless snowfall ratio (prsnratio = prsn/pr):<br><br>([0,1], [0.0001,0.9999]) | 0.5° grid, daily | Derived from bias-adjusted and downscaled pr and prsnratio from GFDL-ESM4, IPSL-CM6A-LR, MPI-ESM1-2-HR, MRI-ESM2-0, and UKESM1-0-LL simulations generated for CMIP6. |
| Surface Air Pressure | **ps** | Pa ([480, 110000]) | 0.5° grid, daily | Bias-adjusted and downscaled from GFDL-ESM4, IPSL-CM6A-LR, MPI-ESM1-2-HR, MRI-ESM2-0, and UKESM1-0-LL simulations generated for CMIP6. |
| Surface Downwelling Longwave Radiation | **rlds** | W m-2 ([40, 600]) | 0.5° grid, daily | Bias-adjusted and downscaled from GFDL-ESM4, IPSL-CM6A-LR, MPI-ESM1-2-HR, MRI-ESM2-0, and UKESM1-0-LL simulations generated for CMIP6. |



| Surface Downwelling Shortwave Radiation | **rsds** | W m-2 ([0, 500])<br><br>Maximum range and inner bounds of normalized rsds used during bias adjustment: ([0,1], [0.0001, 0.9999]) | 0.5° grid, daily | Bias-adjusted and downscaled from GFDL-ESM4, IPSL-CM6A-LR, MPI-ESM1-2-HR, MRI-ESM2-0, and UKESM1-0-LL simulations generated for CMIP6. |
|---|---|---|---|---|
| Near-Surface Wind Speed | **sfcwind** | m s-1 ([0.1, 50], [0.01,∞[) | 0.5° grid, daily | Bias-adjusted and downscaled from GFDL-ESM4, IPSL-CM6A-LR, MPI-ESM1-2-HR, MRI-ESM2-0, and UKESM1-0-LL simulations generated for CMIP6. |
| Near-Surface Air Temperature | **tas** | K ([183.15, 343.15]) | 0.5° grid, daily | Bias-adjusted and downscaled from GFDL-ESM4, IPSL-CM6A-LR, MPI-ESM1-2-HR, MRI-ESM2-0, and UKESM1-0-LL simulations generated for CMIP6. |
| Daily Maximum Near-Surface Air Temperature | **tasmax** | K ([183.15, 343.15])<br><br>Maximum range and inner bounds considered for tasrange: ([0.01, ∞[, [0.01,∞[)<br><br>Maximum range and inner bounds considered for unitless tasskew: ([0,1], [0.0001,0.9999]) | 0.5° grid, daily | Derived from bias-adjusted and downscaled tasrange = tasmax - tasmin and tasskew = (tas - tasmin) / (tasmax - tasmin) from GFDL-ESM4, IPSL-CM6A-LR, MPI-ESM1-2-HR, MRI-ESM2-0, and UKESM1-0-LL simulations generated for CMIP6. |
| Daily Minimum Near-Surface Air Temperature | **tasmin** | K ([183.15, 343.15])<br><br>Maximum range and inner bounds considered for tasrange: ([0.01, ∞[, [0.01,∞[)<br><br>Maximum range and inner bounds considered for unitless tasskew: ([0,1], [0.0001,0.9999]) | 0.5° grid, daily | Derived from bias-adjusted and downscaled tasrange = tasmax - tasmin and tasskew = (tas - tasmin) / (tasmax - tasmin) from GFDL-ESM4, IPSL-CM6A-LR, MPI-ESM1-2-HR, MRI-ESM2-0, and UKESM1-0-LL simulations generated for CMIP6. |




For the pre-industrial conditions, 500 years of picontrol output data are used and harmonised across
General Circulation Models (GCM) with respect to the time range they cover. This is possible because
picontrol data only carry nominal year labels. We shift the GCM-specific picontrol time ranges listed in
**Table 4** to 1601–2100. For the historical and future climate conditions, we provide input data for
1850–2014 and 2015–2100, respectively, in line with the time ranges covered by the corresponding
CMIP6 experiments. The common time axis is important as the use of the input data should be
harmonised across all sectors. In particular, the year-by-year combination of the pre-industrial
climate-related forcing with the historical direct human forcing should be done in the same way across
all sectors and models.

**Selection of climate models.** To limit the number of mandatory impact simulations and hence lower the
barrier to participation in ISIMIP3b, we provide climate input data for only five selected CMIP6 climate
models. The basic characteristics of the five GCMs are listed in **Table 4.** The models were selected based
on data availability at the selection time (late 2019 to early 2020), performance in the historical period,
structural independence, process representation and equilibrium climate sensitivity (ECS).

To be included in ISIMIP3b, a GCM had to provide daily data for all variables listed in **Table 3** except for
huss (which was derived from hurs, ps and tas, see below), ps if sea level pressure (psl) was available, so
a proxy for ps could be computed based on psl and tas, and sfcwind if zonal and meridional near-surface
wind components (uas, vas) were available, so a proxy for sfcwind could be computed based on uas and
vas. Those daily data had to cover 500 picontrol years and all years of the historical, SSP1-2.6, SSP3-7.0,
and SSP5-8.5. In addition, we favoured GCMs that provided the additional  input data needed for the
tropical cyclone modelling (**Table 5**) and the fisheries and marine ecosystems sector (FishMIP; **Table 10**).

**Table 4:** Characteristics of CMIP6 climate models used in ISIMIP3b. Columns show (from left to right) the
climate model acronym, the horizontal grid size (longitude x latitude) of the original atmospheric output
data, the ensemble member used, the nominal time range covered by the picontrol data used, the
equilibrium climate sensitivity (ECS) according to (Meehl et al., 2020), and the main model reference
paper and the CMIP6 simulation data publications used. For definitions of climate model acronyms and
modelling groups see (Durack, n.d.).

| GCM | Grid size | Member | picontrol | ECS | References |
|---|---|---|---|---|---|
| GFDL-ESM4 | 288 x 180 | r1i1p1f1 | 0001–0500 | 2.6°C | (Dunne et al., 2020; John et al., 2018; Krasting et al., 2018) |
| IPSL-CM6A-LR | 144 x 143 | r1i1p1f1 | 1870–2369 | 4.6°C | (Boucher et al., 2018, 2019, 2020) |
| MPI-ESM1-2-HR | 384 x 192 | r1i1p1f1 | 1850–2349 | 3.0°C | (Jungclaus et al., 2019; Mauritsen et al., 2019; Schupfner et al., 2019) |
| MRI-ESM2-0 | 320 x 160 | r1i1p1f1 | 1850–2349 | 3.2°C | (Yukimoto, Kawai, et al., 2019; Yukimoto, Koshiro, et al., 2019a, 2019b) |



| UKESM1-0-LL | 192 x 144 | r1i1p1f2 | 1960–2459 | 5.3°C | (Good et al., 2019; Sellar et al., 2019; Tang et al., 2019) |
|---|---|---|---|---|---|



According to a skill analysis  (see **Figure 2**), the GCMs ACCESS-CM2, AWI-CM-1-1-MR, CESM2,
CESM2-WACCM, CMCC-ESM2, EC-Earth3-AerChem, GFDL-CM4, GFDL-ESM4, HadGEM3-GC31-LL,
HadGEM3-GC31-MM, MPI-ESM1-2-HR, MPI-ESM1-2-LR, MRI-ESM2-0, NorESM2-MM, SAM0-UNICON,
TaiESM1, and UKESM1-0-LL perform relatively well in reproducing the main historically observed
characteristics of the atmosphere. From that list, only GFDL-ESM4, MPI-ESM1-2-HR, MRI-ESM2-0, and
UKESM1-0-LL provided all required daily data at the time of model selection. Another model that
fulfilled all those data requirements and shows an average performance in the historical period is
IPSL-CM6A-LR. These five GCMs were selected to be used in ISIMIP3b. With the exception of
GFDL-ESM4, these models also provide the data needed for tropical cyclone modelling. GFDL-ESM4 is
the model providing the most comprehensive oceanic bio-geochemical forcings for FishMIP while other
models cover less and partly other oceanic variables (see **Table 16**). Three of the climate models
(GFDL-ESM4, IPSL-CM6A-LR, UKESM1-0-LL) are successors of models already used in ISIMP2b and in the
ISIMIP Fast Track.

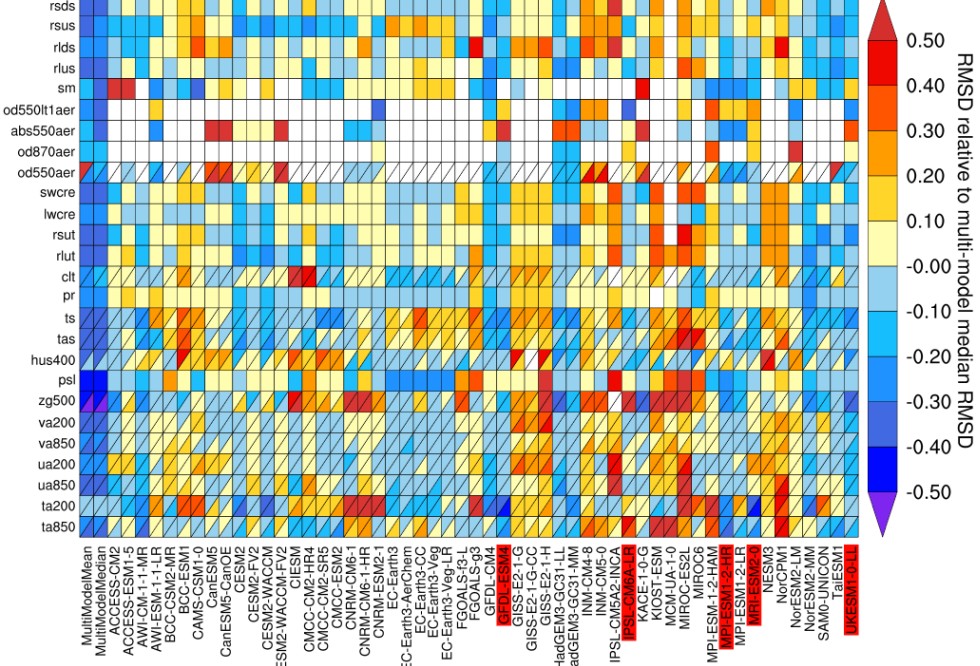



**Figure 2**: Relative space-time root-mean-square deviation (RMSD) calculated from the climatological seasonal
cycle of the CMIP6 historical simulations (1980–1999) compared to observational datasets, for various CMIP6
GCMs (columns) and climate variables (rows), similar to Fig. 6 of (Bock et al., 2020). A relative performance is





displayed, with blue shading being better and red shading worse than the median RMSD of all model results of
the CMIP6 ensemble. A diagonal split of a grid square shows the relative error with respect to the reference data
set (lower right triangle) and an alternative data set (upper left triangle), as listed in Table 5 of (Bock et al., 2020).
White boxes are used when data are not available for a given model and variable. Models selected for ISIMIP3b
are highlighted in red. Variables are (from top to bottom): Surface Downwelling Shortwave Radiation (rsds),
Surface Upwelling Shortwave Radiation (rsus), Surface Downwelling Longwave Radiation (rlds), Surface Upwelling
Longwave Radiation (rlus), Soil Moisture (sm), Ambient Fine Aerosol Optical Depth at 550 nm (od550lt1aer),
Ambient Aerosol Absorption Optical Thickness at 550 nm (abs550aer), Ambient Aerosol Optical Depth at 870 nm
(od870aer), Ambient Aerosol Optical Thickness at 550 nm (od550aer), Shortwave Cloud Radiative Effect (swcre),
Longwave Cloud Radiative Effect (lwcre), Top-of-Atmosphere Outgoing Shortwave Radiation (rsut),
Top-of-Atmosphere Outgoing Longwave Radiation (rlut), Total Cloud Cover Percentage (clt), Precipitation (pr),
Surface Temperature (ts), Near-Surface Air Temperature (tas), Specific Humidity at 400 hPa (hus400), Sea Level
Pressure (psl), Geopotential Height at 500 hPa (zg500), Northward Wind at 200 hPa (va200), Northward Wind at
850 hPa (va850), Eastward Wind at 200 hPa (ua200), Eastward Wind at 850 hPa (ua850), Air Temperature at 200
hPa (ta200), and Air Temperature at 850 hPa (ta850). Produced with ESMValTool v2.0 (Andela, Broetz, de Mora,
Drost, Eyring, et al., 2020; Andela, Broetz, de Mora, Drost, Weigel, et al., 2020; Righi et al., 2020) .


The five GCMs are structurally independent in terms of their ocean and atmosphere model components.
Furthermore, all of them have a coupled climate and carbon cycle and in some cases, fully interactive
chemistry and aerosol components. We favoured models that applied prognostic couplings between
processes and model domains wherever possible to maximise the coverage of simulated feedbacks.

The five GCMs provide a good representation of both the mean and the range of the full CMIP6
multi-model ensemble ECS. According to (Meehl et al., 2020), the CMIP6 multi-model mean ECS is 3.7°C
, which is precisely met by the mean ECS of the five ISIMIP3b GCMs. The transient climate response
(TCR) of 2.0°C is also precisely met. This provides an improvement over ISIMIP2b. In that case the mean
ECS for the full CMIP5 was 3.2°C compared with a mean ECS of 3.72°C for the four ISMIP2b GCMs (see
Table S1 and S2 in (Jägermeyr et al., 2021)). The ISIMIP3b ensemble includes three models with
below-average ECS (GFDL-ESM4, MPI-ESM1-2-HR, MRI-ESM2-0) and two models with above-average
ECS (IPSL-CM6A-LR, UKESM1-0-LL) (see **Table 12**). In line with their ECS values, we find GFDL-ESM4 and
UKESM1-0-LL to project the weakest and strongest global warming, respectively, under any future
scenario considered (see **Figure 3**). Under SSP5-8.5, the global mean near-surface temperature in 2100
is about 3°C larger in UKESM1-0-LL than in GFDL-ESM4. Under SSP1-2.6, the projections are about 1.5°C
apart. The ensemble mean warming of the ISIMIP3b CMIP6 models is significantly higher than the
warming of the ISIMIP2b CMIP5 models, across global land area by an average of 0.3°C, but over the
main breadbasket cropland regions by more than 0.5°C between 1983–2013 and 2069–2099, under
both SSP1-2.6 and SSP5-8.5 (Table S1 in (Jägermeyr et al., 2021).



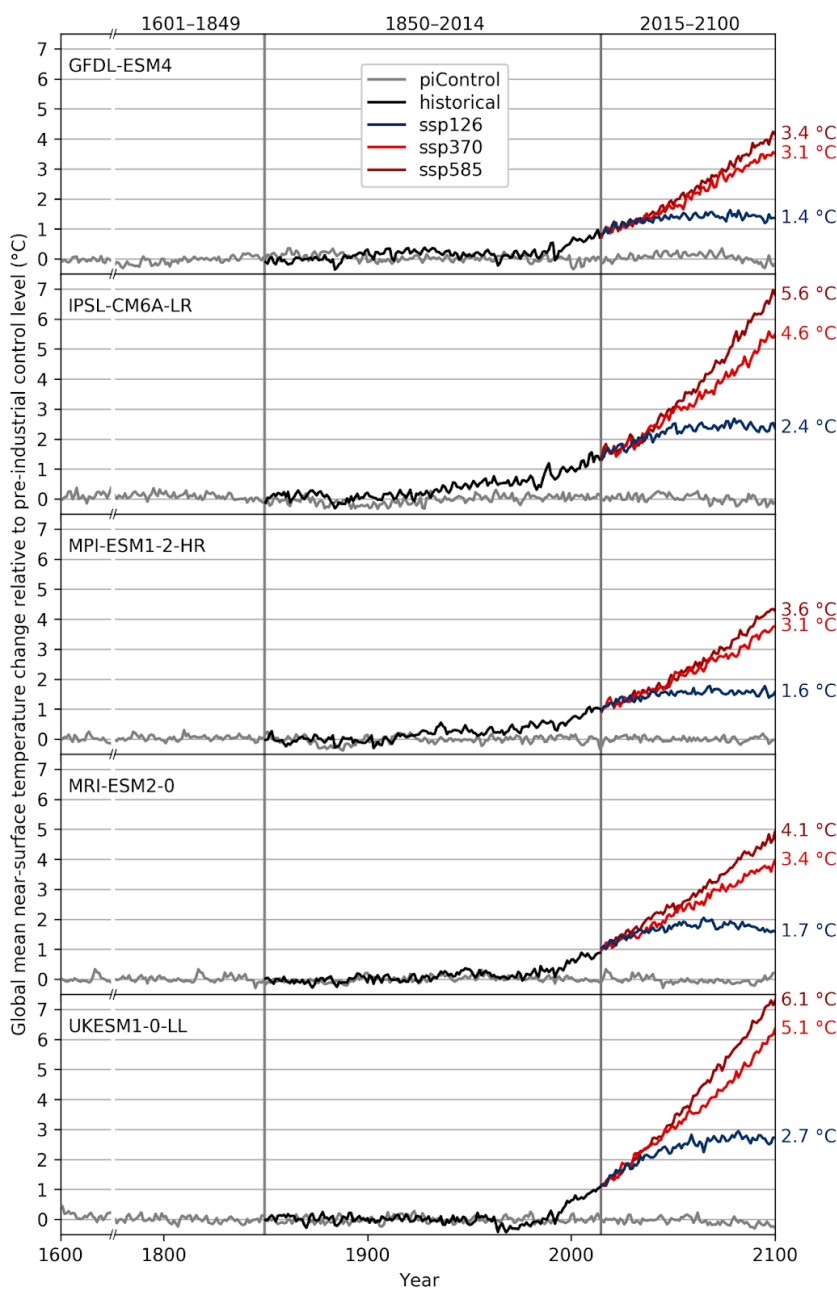

**Figure 3**: Time series of annual global mean near-surface temperature change relative to pre-industrial levels (1601–1849 average) as simulated with GFDL-ESM4, IPSL-CM6A-LR, MPI-ESM1-2-HR, MRI-ESM2-0 and UKESM1-0-LL (from top to bottom). Colour coding indicates the underlying CMIP6 experiments (grey: pre-industrial control, black: historical, blue: SSP1-2.6, light red: SSP3-7.0, dark red: SSP5-8.5) with corresponding time periods given at the top. Numbers to the right of the plot represent end-of-century warming levels under the






different future scenarios, expressed as the global multi-year mean near-surface temperature change from
1601–1849 to 2070-2100.

**Bias adjustment and statistical downscaling.** To make the GCM-based climate forcing usable for the
impact modellers we apply a bias adjustment ensuring that the GCM simulations match the observed
distribution of climate data over the historical reference period (1979–2014). In addition to the bias
adjustment a statistical downscaling to our standard 0.5° grid is included in the pre-processing of the
surface and near-surface atmospheric variables (see **Table 11**). The method used for the bias adjustment
and statistical downscaling (BASD) in ISIMIP3b is version 2.5 of ISIMIP3BASD  (Lange, 2019b, 2021a).

ISIMP3BASD has several advantages compared to the method used for bias adjustment and statistical
downscaling in ISIMIP2b (Frieler et al., 2017; Lange, 2017, 2018). First, it clearly separates the
adjustment of biases in climate model output at 1° or 2° resolution, whatever is closest to the original
output data, from the statistical downscaling to the target resolution of 0.5°. Compared to ISIMIP2b,
where climate model output was first spatially interpolated to the target resolution and then
bias-adjusted, the new approach improves the spatial variability at the target resolution (Lange, 2019b).
Second, the new quantile mapping method preserves trends in each quantile of the distribution of the
daily data and adjusts biases in distribution quantiles of the daily data more accurately than the
ISIMIP2b bias adjustment methods (Lange, 2019b).

For trend preservation, we first produce pseudo-future observations by shifting the historically observed
daily data by the simulated future climate change. Here, the signal of climate change is the difference or
the ratio between the inverse empirical cumulative distribution function of the historical period and the
respective distribution functions of each 36-year period of the future. Using the difference ensures
additive trend preservation and using the ratio ensures multiplicative trend preservation under bias
adjustment. We apply additive trend preservation for near-surface air temperature (tas), sea level
pressure (psl, see **Table 6**), and surface downwelling longwave radiation (rlds). We apply primarily
multiplicative trend preservation for precipitation including snowfall (pr), near-surface wind speed
(sfcWind), and the range (tasrange = tasmax - tasmin) between the daily maximum and minimum
near-surface air temperatures (tasmax and tasmin, respectively) that can transition smoothly to additive
trend preservation for data with large negative biases in the historical period (Lange, 2019b). In a second
step, the future simulations are mapped onto the pseudo-future observations by quantile mapping.
Both steps, the generation of the pseudo future observations and the quantile mapping of the future
simulations onto the pseudo observations, are applied for each day of the year separately. The
distributions include data from the 31 days around the considered day and all years of the reference or
future period, respectively. This means a sample size of 31x36 values for each day of the year. Through
this approach the bias adjustment implicitly also adjusts the multi-year mean annual cycle and a mix of
year-to-year and day-to-day variability (Haerter et al., 2011).

In addition, the method adjusts the frequency of daily data falling outside of the inner bounds specified
in **Table 11** (e.g. the dry day frequency, i.e. the number of days with precipitation below 0.1 mm day-1).



Four variables were adjusted and downscaled indirectly: near-surface specific humidity (huss) was
derived from adjusted and downscaled near-surface relative humidity (hurs), surface air pressure (ps),
and near-surface air temperature (tas) using the equations of (Buck, 1981) as described in (Weedon et
al., 2010), snowfall (prsn) was derived from adjusted and downscaled precipitation including snow (pr)
and the snowfall ratio (prsnratio = prsn / pr), and daily maximum and daily minimum near-surface air
temperatures (tasmax and tasmin, respectively) were derived from adjusted and downscaled tas, and
the tasrange = tasmax - tasmin and skewness of the daily temperature cycle tasskew = (tas - tasmin) /
(tasmax - tasmin).

The basic characteristics of ISIMIP3BASD (version 1.0) are described in (Lange, 2019b). However, the
method finally used to generate the forcing data now provided within ISIMIP3b (ISIMIP3BASD version
2.5) deviates from the original version in some aspects. In the following we describe the most important
updates of the procedure relative to the one described in (Lange, 2019b). For a complete list of
differences between the two versions of the BASD method and the full history of which feature was
added in which update, see the CHANGELOG included in the archive of code version 2.5  (Lange, 2021a).

In (Lange, 2019b) the bias-adjustment was applied on a monthly basis, i.e. the pseudo-future
observations and the quantile mapping described above was applied to all daily January data, February
data and so forth. This approach can introduce discontinuities at the transition from one month to
another (see **Figure 4**). That is why for ISIMIP3b the adjustment is done in the running window mode
with steps of one day and a window width of 31 days as described above. This approach resolves the
discontinuity issue (see **Figure 4**), as suggested by (Themeßl et al., 2012); (Thrasher et al., 2012);
(Gennaretti et al., 2015); and (Grenier, 2018).

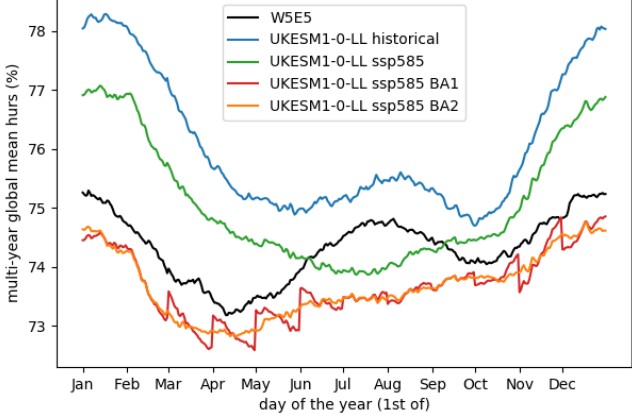


**Figure 4:** Global multi-year daily mean near-surface relative humidity for UKESM1-0-LL historical (1979-2014) and
SSP5-8.5 (2065-2100), with historical simulated data in blue, future simulated data in green, future bias-adjusted
data in red and orange, and observational reference data in black. A smooth annual cycle is produced if
ISIMIP3BASD v2.5 is applied in running-window mode in steps of one day (orange, BA2). In contrast, a
month-by-month application, which was the only option in ISIMIP3BASD v1.0, generates discontinuities at each
turn of the month (red, BA1).





Since ps, rlds and tas can show significant trends within the 36-year training and application periods
ISIMIP3BASD v1.0 includes a detrending of these variables within these intervals before the pseudo
future observations and the transfer functions are estimated and applied. Afterwards the trend is added
back again. This is done to prevent the confusion of trends with interannual variability during quantile
mapping (Lange, 2019b; Maraun, 2013). In contrast to v1.0, in v2.5, applied to generate the ISIMIP3b
forcings data, the detrending is only applied if the trend is significantly different from zero at the 5%
level.

We also changed the method used to generate future pseudo-observations of bounded variables
(equations (8) and (9) of (Lange, 2019b)), in order to stabilise results in some edge cases. If, e.g., the
historically observed relative dry-day frequency was 0.0 while the simulated frequency was 0.8 for the
historical period and 0.9 for some future period, then, according to equation (9) of (Lange, 2019b), the
future pseudo-observed frequency would be equal to $1 - (1 - 0.0)(1 - 0.9)/(1 - 0.8) = 0.5$. As
this is considered unrealistic we apply a revised version of equation (9) of (Lange, 2019b) that reads

$$
\quad P^{obs}_{fut} = \{
$$

$$
\quad P^{sim}_{fut} \text{ if } P^{sim}_{hist} = P^{obs}_{hist},
$$

$$
\quad 0 + (P^{obs}_{hist} - 0)(P^{sim}_{fut} - 0)/(P^{sim}_{hist} - 0) \text{ if } P^{sim}_{fut} \leq P^{sim}_{hist} > P^{obs}_{hist},
$$

$$
\quad 1 - (1 - P^{obs}_{hist})(1 - P^{sim}_{fut})/(1 - P^{sim}_{hist}) \text{ if } P^{sim}_{fut} \geq P^{sim}_{hist} < P^{obs}_{hist},
$$

$$
\quad P^{obs}_{hist} + P^{sim}_{fut} - P^{sim}_{hist} \text{ otherwise.} \tag{1}
$$


In this revised relation, the otherwise case applies if $P^{sim}_{fut} < P^{sim}_{hist} < P^{obs}_{hist}$ or
$P^{sim}_{fut} > P^{sim}_{hist} > P^{obs}_{hist}$. Hence it applies to the aforementioned edge case, where it produces a less
extreme future pseudo-observed relative frequency of $0.0 + 0.9 - 0.8 = 0.1$. Equation (8) of
(Lange, 2019b) was revised analogously to equation (9).

Furthermore, we refined the method used to generate future pseudo-observations (step 5 of the bias
adjustment algorithm of (Lange, 2019b)) for all variables with at least one bound: In v1.0, the future
pseudo observations were generated by transferring simulated trends in all distribution quantiles to the
observational reference data. That included trends in, e.g., precipitation quantiles below the wet-day
threshold. However, in some cases, the trend transfer turned many dry days into wet days, with a
profound impact on the shape of the distribution of future pseudo-observed wet-day precipitation. As a
result, simulated trends in wet-day precipitation intensity were not well preserved. In v2.5, trend
transfers are restricted to values within threshold. This particularly improves the preservation of trends
in wet-day precipitation intensities.



We also modified the bias adjustment method for Near-Surface Relative Humidity (hurs) because
ISIMIP3BASD v1.0 turned out to produce unrealistic distributions of hurs under climate change if there
are too many cases of supersaturation (hurs ≥ 100%) in the simulated data. This is the case for several
of the CMIP6 GCMs selected for ISIMIP3b, particularly in high-latitude winter: While no supersaturations
are found in the observational reference data, the GCM simulates many supersaturations in the
historical reference period and even more so in a future period, under SSP5-8.5 (see **Figure 5**).
ISIMIP3BASD v1.0 preserves this projected trend and hence produces future bias-adjusted hurs data
with many supersaturations. In v2.5, this trend is no longer preserved. Instead, the supersaturation
probability is fixed at the observed level, which is zero or very close to zero in all seasons and grid cells
for W5E5. Future pseudo observations of hurs are generated by applying the revised (see above)
equation (8) of (Lange, 2019b) to all hurs values after capping them at 100%. The new approach was
motivated by findings from (Ruosteenoja et al., 2017, 2018). They analysed hurs data from CMIP5 and
showed that (i) supersaturations in those data are mostly spurious, resulting from, e.g., inconsistencies
in the interpolation of temperature and specific humidity to the near-surface level, and (ii) climatological
mean value trends of hurs become more consistent with trends in relative humidity from the lowest
model level if hurs is capped at 100% before trends are calculated.

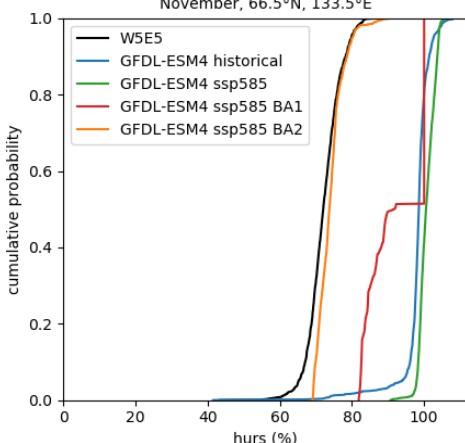



**Figure 5:** Empirical cumulative distribution functions of near-surface relative humidity in high-latitude winter
(November, 66.5°N, 133.5°E) for GFDL-ESM4 historical (1979-2014) and SSP5-8.5 (2065-2100), with historical
simulated data in blue, future simulated data in green, future bias-adjusted data in red and orange, and
observational reference data in black. The simulated climate change signal is well preserved with ISIMIP3BASD
v2.5 using a fixed supersaturation (hurs ≥ 100%) probability and equation (1) applied to all hurs values after
capping them at 100% to generate future pseudo observations (orange, BA2). In contrast, the simulated climate
change signal is not well preserved if the supersaturation probability is allowed to change and equations (8) and
(9) of (Lange, 2019b) are used to generate future pseudo observations of hurs (red, BA1).

In addition, while ISIMIP3BASD v1.0 applies parametric quantile mapping to all climate variables, we
used a nonparametric approach for the bias adjustment of near-surface relative humidity (hurs), the



snowfall ratio (prsnratio), surface downwelling shortwave radiation (rsds), and the skewness of the daily
temperature (tasskew) since the parametric quantile mapping method previously used for those
variables suffered from occasionally unstable beta distribution fits.

Moreover, the parametric quantile mapping described in (Lange, 2019b) does not only adjust biases in
quantiles of the simulated daily data but also adjusts biases in the likelihood of individual events, as in
(Switanek et al., 2017). To avoid overfitting artefacts we did not adjust event likelihoods for ISIMIP3b.

Finally, the diurnal temperature range (tasrange) was ultimately bias-adjusted using a Weibull
distribution, not a Rice distribution as described in (Lange, 2019b) because the Weibull distribution fits
the data better in most cases, in particular in the upper tail.

For further details of the application of ISIMIP3BASD v2.5 for ISIMIP3b, including the exact Python
commands and application periods used per CMIP6 experiment, see the ISIMIP3b bias adjustment fact
sheet (Lange, 2021b).

In addition, we use a new observational target dataset. Instead of using the EWEMBI dataset (E2OBS,
WFDEI and ERAI data merged and bias-corrected for ISIMIP; (Lange, 2019a) in ISIMIP3b we adjust the
climate forcing data to version 2.0 of the W5E5 dataset (WFDE5 over land merged with ERA5 over the
ocean; (Lange et al., 2021). The data cover the entire globe at 0.5° horizontal and daily temporal
resolution from 1979 to 2019. W5E5 v2.0 is derived by applying version 2.0 of the WATCH Forcing Data
methodology (WFDE5; (Cucchi et al., 2020) to ERA5 reanalysis data (Hersbach et al., 2020) and
precipitation data from version 2.3 of the Global Precipitation Climatology Project (GPCP; (Adler et al.,
696 2003)).


The statistical downscaling method did not change between v1.0 and v2.5 of ISIMIP3BASD, i.e. for
ISIMIP3b we use the approach described (Lange, 2019b). This method adds the spatiotemporal
variability that is missing at the low spatial resolution at which the bias adjustment is done (1° or 2°,
depending on the GCM), compared to the target resolution of the downscaling (0.5°). The method is a
modified version of the MBCn algorithm from (Cannon, 2018), which in turn is a stochastic, multivariate,
non-parametric quantile mapping method. We use it to transfer the statistical relationship between
low-resolution and high-resolution W5E5 data to the GCM output that was previously bias-adjusted
using low-resolution W5E5 data. In comparison to the approach used in ISIMIP2b (a spatial interpolation
to the target resolution followed by a bias adjustment at that resolution), the approach used in ISIMIP3b
is less prone to inflate temporal variability and deflate spatial variability, i.e. the ISIMIP3b approach
produces more realistic spatiotemporal variability patterns at the target resolution  (Lange, 2019b).

**2.2 Tropical cyclones**

**Table 5:** Information about tropical cyclone tracks and windfields provided as climate-related forcing
data within ISIMIP3b.



| Variable | Variable specifier | Unit | Resolution | Datasets |
|---|---|---|---|---|
| Time associated with a given location of the storm centre | **time** | hours since 1950-01-01 00:00 | along-track, 2-hourly (MIT model) 6-hourly (CHAZ model) | MIT (Emanuel et al., 2008) and CHAZ (Lee et al., 2018) |
| Latitudinal coordinate of storm centre | **lat** | degrees north | along-track, 2-hourly (MIT model) 6-hourly (CHAZ model) | MIT (Emanuel et al., 2008) and CHAZ (Lee et al., 2018) |
| Longitudinal coordinate of storm centre | **lon** | degrees east | along-track, 2-hourly (MIT model) 6-hourly (CHAZ model) | MIT (Emanuel et al., 2008) and CHAZ (Lee et al., 2018) |
| Central pressure | **pres** | hPa | along-track, 2-hourly | MIT (Emanuel et al., 2008) |
| Maximum 1-minute sustained wind speed | **windspatialmax** | knots | along-track, 2-hourly (MIT model) 6-hourly (CHAZ model) | MIT (Emanuel et al., 2008) and CHAZ (Lee et al., 2018) |
| Radius of maximum wind speeds | **rmw** | km | along-track, 2-hourly | MIT (Emanuel et al., 2008) |
| Wind speed on the 850 hPa pressure level | **u850 v850** | knots (MIT model), ms$^{-1}$ (CHAZ model) | along-track, 2-hourly (MIT model) 6-hourly (CHAZ model) | MIT (Emanuel et al., 2008) and CHAZ (Lee et al., 2018) |
| Temperature on the 600 hPa pressure level | **t600** | K | along-track, 2-hourly (MIT model) 6-hourly (CHAZ model) | MIT (Emanuel et al., 2008) and CHAZ (Lee et al., 2018) |
| Frequency of TC occurrence | **freqyear** | count per year | annual | MIT (Emanuel et al., 2008) |
| Gridded lifetime maximum 1-minute sustained wind speed | **windlifetimemax** | ms$^{-1}$ | Per storm on a 300 arc-seconds (~10 km) grid | Wind fields calculated with Holland and Emanuel-Rotunno wind profiles (Holland, 1980, 2008) for both sets of synthetic tracks (CHAZ and MIT) |
| Maximum 24-hourly rainfall total during the whole storm duration | **maxrain** | mm | per storm on a 300 arc-seconds (~10 km) grid | Maximum 24-hourly rainfall (Zhu et al., 2013) calculated for Holland and Emanuel-Rotunno wind profiles for both sets of synthetic tracks (CHAZ and MIT) |




We provide large ensembles of potential realisations of TC tracks and intensities that are consistent with
the large-scale atmospheric and oceanic conditions simulated by the 5 ISIMIP3b GCMs (see **Table 4**) and
for a selection of scenarios considered in ISIMIP3b (see **Table 1**). We provide gridded wind (maximum
1-minute sustained wind speeds during the whole duration of the TC) and rainfall (maximum 24-hourly
amounts of rain during the whole duration of the TC) fields at a spatial resolution of 300 arc-seconds
(approximately 10 km) by the same approaches also applied to the historically observed tracks ((Frieler
et al., 2024), section **3.2**).
The tracks are generated by two different statistical-dynamical approaches that, forced by GCM data
(see **Table 4**), generate a large number of synthetic storms. Both methods to generate the TC tracks
consist of a genesis, a track, and an intensity module:
**The MIT approach.** Within MIT (Emanuel et al., 2008), the time-evolving state of the atmosphere and
ocean surface given by the GCMs is randomly (uniformly distributed in time and space) seeded by weak
proto-cyclones (genesis module). The seed disturbances are assumed to move with the GCM-provided
large-scale flow in which they are embedded, plus a westward and poleward component owing to
planetary curvature and rotation (track module). Their intensity is calculated using the Coupled
Hurricane Intensity Prediction System (CHIPS; (Emanuel et al., 2004), a simple axisymmetric hurricane
model coupled to a reduced upper ocean model to account for the effects of upper ocean mixing of cold
water to the surface (intensity module). Applied to the synthetically generated tracks, this model
simulates which of the seeded proto-cyclones develop into TCs, reaching maximum 1-minute sustained
wind speeds of at least 35 knots, or dissipate due to unfavourable environments. The probabilistic
seeding of proto-cyclones is repeated until the desired number of storms per year is reached (in our
case, 1500). For each year, the share of proto-cyclones that dissipated in the process is used to derive an
estimate of annual TC occurrences (**freqyear**). Extensive comparisons to historical events (Emanuel et
al., 2008) have revealed that the statistical properties of the simulated events are consistent with
historical TC genesis.
1500 tracks were generated globally and for each year of the ISIMIP3b period 1850—2100 (except for
GFDL-ESM4, where tracks were only generated for 1850-2014 and 2061-2100, and MRI-ESM2-0 for
1950-2100, see **Table 1**). Depending on the application, a simple subsampling (Meiler et al., 2022) or a
more advanced bias-correction and emulation procedure (Geiger et al., 2021) might be necessary to
extract properly-sized sets of potential realisations from the MIT ensembles.
The MIT track data shall be used for non-commercial research or academic purposes only. Data can be
made available by the ISIMIP team upon written consent by Kerry Emanuel (MIT, email:
emanuel@mit.edu).
**The CHAZ approach.** CHAZ (Lee et al., 2018) seed disturbances are also initialised randomly, but, in
contrast  to the MIT model, the global seeding rate and the local probabilities are derived from two
versions of a TC genesis index (TCGI, (Tippett et al., 2011) (genesis module) and intended to represent
the environmental conditions instead of being adjusted to produce a prescribed number of TCs. It is
noted that CHAZ's projection of global and basin-wide TC annual frequency is sensitive to the choice of
the particular variable used to represent moisture in its genesis module. Simulations using column




relative humidity (CRH) as the moisture variable tend to project an overall increase in global TC frequency, while those using saturation deficit (SD) show a decrease (Camargo et al., 2014), (Lee et al., 2020). Both parameters describe how far the atmosphere is from saturation, and they have very similar spatial patterns in the present climate, so historical data cannot be used to determine which variable is the best choice to represent the climate. These two configurations reflect the uncertainty of TC frequency projections (Sobel et al., 2021). Here we provide CHAZ downscaling using both choices of moisture variable to account for this uncertainty.

Similar to MIT, CHAZ then moves the synthetic storms by advection of the environmental steering flow plus a beta drift (track module). The evolution of synthetic storm intensity is calculated using the surrounding atmospheric conditions through an empirical multiple linear regression model plus a stochastic component (intensity module, (Lee et al., 2015, 2016)). The stochastic component accounts for internal storm dynamics that do not depend explicitly on the environment. While, in MIT, TC occurrence frequency is provided as an additional variable, in CHAZ, this information is implicitly contained in the number of TCs that were seeded by the genesis module and that reached TC strength according to the intensity module.

For ISIMIP3b, 20 different CHAZ realisations of the genesis and subsequent tracks are generated with 40 ensemble members each from the intensity module. For each of the 20 realisations, we compute wind and rain fields for the first ensemble member from the intensity ensemble. The design of 20 realisations allows CHAZ to generate similar numbers (~1800) of synthetic storms per year per GCM as the MIT models over the historical period. The exact number of storms per year in CHAZ varies by GCM, by scenario, by the choice of humidity variables in CHAZ's genesis component (Lee et al., 2020). On average, CHAZ generates 1817, 1802, 1820, 1810, 1842 storms per year for GFDL-ESM4, IPSL-CM6A-LR, MPI-ESM1-2-HR, MRI-ESM2-0, and UKESM1-0-LL, respectively. The CHAZ model has been shown to capture the statistical properties of the observed storms when forced by a global reanalysis data (Lee et al., 2018). Its CMIP6 downscaling results are reported in (Fosu et al., 2024). (Sobel et al., 2019) used both models to study cyclone risk at Mumbai, India and showed that MIT and CHAZ generate comparable return periods (frequency of exceedance) of maximum wind speeds at landfall. However, a frequency bias-correction might still be necessary, depending on the application (Meiler et al., 2022).

The track data generated by the CHAZ approach shall be used for non-commercial research or academic purposes only. Data can be made available by the ISIMIP team upon written consent by Chia-Ying Lee (Columbia University, email: cl3225@columbia.edu).

**Table 6:** Climate input data interpolated to 2° horizontal resolution and provided without bias adjustment for tropical cyclone modelling with MIT and CHAZ.

| Variable | Variable specifier | Unit | Resolution | Datasets |
|---|---|---|---|---|
| Sea Water Potential Temperature | **thetao** | °C | 2° grid, model specific levels (m from surface to 200m depth), monthly | IPSL-CM6A-LR, MPI-ESM1-2-HR, MRI-ESM2-0, and UKESM1-0-LL simulations generated for CMIP6. |



| Sea Surface Temperature | **tos** | °C | 2° grid over the ocean, monthly | IPSL-CM6A-LR, MPI-ESM1-2-HR, MRI-ESM2-0, and UKESM1-0-LL simulations generated for CMIP6. |
|---|---|---|---|---|
| Surface Temperature | **ts** | K | 2° grid covering land and ocean areas, monthly | IPSL-CM6A-LR, MPI-ESM1-2-HR, MRI-ESM2-0, and UKESM1-0-LL simulations generated for CMIP6.<br><br>ts may differ from tos in regions of sea ice where tos refers to temperatures under the ice while ts refers to temperatures at the surface. |
| Air Temperature | **ta** | K | 2° grid; 15 pressure levels (from 1000 to 30 hPa), monthly | IPSL-CM6A-LR, MPI-ESM1-2-HR, MRI-ESM2-0, and UKESM1-0-LL simulations generated for CMIP6. |
| Specific Humidity | **hus** | kg kg-1 | 2° grid; 15 pressure levels (from 1000 to 30 hPa), monthly | IPSL-CM6A-LR, MPI-ESM1-2-HR, MRI-ESM2-0, and UKESM1-0-LL simulations generated for CMIP6. |
| Relative Humidity at 600 hPa | **hur** | % | 2° grid, monthly | IPSL-CM6A-LR, MPI-ESM1-2-HR, MRI-ESM2-0, and UKESM1-0-LL simulations generated for CMIP6. |
| Precipitable water (water vapour content vertically integrated through the atmospheric column) | **prw** | kg m-2 | 2° grid, monthly | IPSL-CM6A-LR, MPI-ESM1-2-HR, MRI-ESM2-0, and UKESM1-0-LL simulations generated for CMIP6. |
| Sea Level Pressure | **psl** | Pa | 2° grid, monthly | IPSL-CM6A-LR, MPI-ESM1-2-HR, MRI-ESM2-0, and UKESM1-0-LL simulations generated for CMIP6. |
| Eastward Wind | **ua** | m s-1 | 2° grid; 200, 250, 850 hPa; monthly | IPSL-CM6A-LR, MPI-ESM1-2-HR, MRI-ESM2-0, and UKESM1-0-LL simulations generated for CMIP6. |
| Northward Wind | **va** | m s-1 | 2° grid; 200, 250, 850 hPa; monthly | IPSL-CM6A-LR, MPI-ESM1-2-HR, MRI-ESM2-0, and UKESM1-0-LL simulations generated for CMIP6. |
| Eastward Wind | **ua** | m s-1 | 2° grid; 250, 850 hPa; daily | IPSL-CM6A-LR, MPI-ESM1-2-HR, MRI-ESM2-0, and UKESM1-0-LL simulations generated for CMIP6. |



| Northward Wind | **va** | m s-1 | 2° grid; 250, 850 hPa; daily | IPSL-CM6A-LR, MPI-ESM1-2-HR, MRI-ESM2-0, and UKESM1-0-LL simulations generated for CMIP6. |
| --- | --- | --- | --- | --- |

## 2.3 Coastal water levels

**Table 7:** Coastal water level specifications

| Variable | Variable specifier | Unit | Resolution | Datasets |
| --- | --- | --- | --- | --- |
| Coastal water levels | **cwl** | m | custom coastal grid<br><br>Hourly or daily maxima | planned |

We do not yet provide coastal water levels as forcing data for ISIMIP3b. However, we plan to generate time series of coastal water levels from 1900 to 2100 at hourly resolution or for daily maxima. The data set and method will be described in a separate manuscript. Similar to the hourly water level dataset of ISIMIP3a (see section **3.3** of (Frieler et al., 2024) and (Treu et al., 2023)), we will combine longer-term annual sea level change with estimates of short-term coastal water level variation. Concerning the long-term sea level change component, we will further develop the ISIMIP2b approach (Frieler et al., 2017) and use tide gauge, satellite, vertical land motion and global climate model data to constrain a model with observations and IPCC AR6 projections in a Bayesian setting. Modelled global contributions from ice sheets and fingerprints are translated to regional sea level rise via fingerprints. A new aspect is that we include an estimation of vertical land motion to provide relative coastal water levels instead of geocentric coastal water levels. A version of the model that projects sea level rise at tide gauge stations (and not all coastlines as is needed here) is currently in review ( (Perrette & Mengel, submitted 2024) ).

We plan to estimate the short-term coastal water level variation by a machine-learning approach that is trained to reproduce simulations of the Global Surge and Tide Model (GTSM) model driven by ERA5 reanalysis data (Muis et al., 2020) or simulations from HighResMIP (Muis et al., 2023). We are currently testing the dependency of the short-term water level variation on available atmospheric information at GCM resolution. If the predictive power is high enough we will use the findings to provide computationally efficient water level projections specific for the ISIMIP GCMs.

## 2.4 Ocean data

In the default experiments, the ocean variables provided by the GCMs are not subject to bias-adjustment, unlike the atmospheric forcing data (section **2.4.1**). This is due to the absence of a comprehensive global observational oceanic dataset to serve as a reference for the adjustment.



However, in order to mitigate potential biases in global impact model simulations stemming from biases
in raw oceanic forcing data, we provide a de-biased version to be used in a sensitivity experiment (see
**Table 2**). They will be derived from an ocean-biogeochemistry model forced by bias-adjusted monthly
atmospheric surface flux data from four of the five ISIMIP3b GCMs. The approach preserves the monthly
variability of the underlying GCM while the daily variability is added from an independent simulation
(see section **2.4.2**).
For the regional impact model simulations, observational data for individual variables have either been
applied directly (if the required forcing was observed) to rectify biases in regional oceanic forcings by
the delta method or have first been translated into the required forcing variable by model simulations
(see section **2.4.3**). In the delta approach absolute simulated deviations from reference levels are added
to the observed reference levels. The regional bias-adjustment is independent from the generation of
the global de-biased forcing data.
In order to gauge the effects of these adjustments on the corresponding impact simulations, the
protocol includes sensitivity experiments (**'de-biased'**) grounded on these adjusted climate-related
forcings (see **Table 2**). The comparison of associated impact simulation to the default ones is expected
to provide valuable insights into the effects of potential biases in the climate-related forcings. The
'de-biased' experiments are considered a starting point to develop methods to bias-adjust the oceanic
forcings in further ISIMIP simulation rounds and make these simulations the default ones. Following the
ISIMIP 'consistency framing' the bias-adjustment should also preserve the daily variability of the original
GCM simulations to allow for a cross-sectoral integration on daily time scale.  .
**2.4.1 Raw data without bias adjustment (default experiment)**
In ISIMIP3b, a  set of physical and biogeochemical ocean variables nearly identical to that in ISIMIP3a is
provided (see section **3.4**, **Table 8** of (Frieler et al., 2023) and **Table 8** below). These variables are
obtained from the CMIP6 GCMs, which also supply the atmospheric forcing for ISIMIP3b, except for
MRI-ESM2-0, which lacks bio-geochemical variables. In other models, only certain individual variables
are missing (see **Table 8**). Obtaining both atmospheric and oceanic variables from the same set of GCMs
ensures consistency between the fisheries and marine ecosystems sector and other ISIMIP sectors. The
available variables in ISIMIP3b are interpolated from the native grids of the ocean models to a regular 1°
grid. This resolution is comparatively lower than that of the ISIMIP3a ocean input data due to the
generally reduced native resolution of CMIP6 GCM simulations compared to the ocean model used to
generate the oceanic forcings based on observational atmospheric forcings for ISIMIP3a.

**Table 8:** Oceanic climate-related forcing data provided within ISIMIP3b. Variables with suffixes -bot,
-surf, and -vint were obtained from the seafloor, the top layer of the ocean, and vertical integration,
respectively.

| Variable | Variable specifier | Unit | Resolution | Datasets |
|---|---|---|---|---|
|  |  |  |  |  |



| Mass concentration of total phytoplankton expressed as chlorophyll | **chl** | kg m-3 | 1° grid, vertically resolved, monthly | GFDL-ESM4, IPSL-CM6A-LR, MPI-ESM1-2-HR, UKESM1-0-LL |
|---|---|---|---|---|
| Sea floor depth | **deptho** | m | 1° grid, constant | GFDL-ESM4, IPSL-CM6A-LR, MPI-ESM1-2-HR, UKESM1-0-LL |
| Downward flux of particulate organic carbon | **expc-bot** | mol m-2 s-1 | 1° grid, monthly | GFDL-ESM4, IPSL-CM6A-LR, MPI-ESM1-2-HR, UKESM1-0-LL |
| Particulate organic carbon content | **intpoc** | kg m-2 | 1° grid, monthly | GFDL-ESM4, MPI-ESM1-2-HR, UKESM1-0-LL |
| Net primary organic carbon production by all types of phytoplankton | **intpp** | mol m-2 s-1 | 1° grid, monthly | GFDL-ESM4, IPSL-CM6A-LR, MPI-ESM1-2-HR, UKESM1-0-LL |
| Net primary organic carbon production by diatoms | **intppdiat** | mol m-2 s-1 | 1° grid, monthly | GFDL-ESM4, IPSL-CM6A-LR, UKESM1-0-LL |
| Net Primary Organic Carbon Production by Other Phytoplankton | **intppmisc** | mol m-2 s-1 | 1° grid, monthly | GFDL-ESM4, IPSL-CM6A-LR, UKESM1-0-LL |
| Net Primary Mole Productivity of Carbon by Picophytoplankton | **intpppico** | mol m-2 s-1 | 1° grid, monthly | GFDL-ESM4 |
| Net Primary Organic Carbon Production of Carbon by Diazotrophs | **intppdiaz** | mol m-2 s-1 | 1° grid, monthly | GFDL-ESM4, MPI-ESM1-2-HR |
| Mixed layer depth defined by delta rho = 0.125 | **mlotstmax** | m | 1° grid, monthly | IPSL-CM6A-LR, MPI-ESM1-2-HR, UKESM1-0-LL |
| Dissolved oxygen concentration | **o2, o2-bot, o2-surf** | mol m-3 | 1° grid, vertically resolved, ocean bottom and surface fields, monthly | GFDL-ESM4, IPSL-CM6A-LR, MPI-ESM1-2-HR, UKESM1-0-LL |



| pH | **ph, ph-bot, ph-surf** | 1 | 1° grid, vertically resolved, ocean bottom and surface fields, monthly | GFDL-ESM4, IPSL-CM6A-LR, MPI-ESM1-2-HR, UKESM1-0-LL |
|---|---|---|---|---|
| Total phytoplankton carbon concentration | **phyc, phyc-vint** | mol m-3 | 1° grid, vertically resolved and vertically integrated, monthly | GFDL-ESM4, IPSL-CM6A-LR, MPI-ESM1-2-HR, UKESM1-0-LL |
| Concentration of diatoms expressed as carbon in sea water | **phydiat, phydiat-vint** | mol m-3 | 1° grid, vertically resolved and vertically integrated, monthly | GFDL-ESM4, IPSL-CM6A-LR, UKESM1-0-LL |
| Concentration of diazotrophs expressed as carbon in Sea Water | **phydiaz, phydiaz-vint** | mol m-3 | 1° grid, vertically resolved and vertically integrated, monthly | GFDL-ESM4, MPI-ESM1-2-HR |
| Mole Content of Miscellaneous Phytoplankton Expressed as Carbon in Sea Water | **phymisc, phymisc-vint** | mol m-2 | 1° grid, vertically resolved and vertically integrated, monthly | GFDL-ESM4, IPSL-CM6A-LR, MPI-ESM1-2-HR, UKESM1-0-LL |
| Mole Concentration of Picophytoplankton Expressed as Carbon in Sea Water | **phypico, phypico-vint** | mol m-3 | 1° grid, vertically resolved and vertically integrated, monthly | GFDL-ESM4 |
| Net Downward Shortwave Radiation at Sea Water Surface | **rsndts** | W m-2 | 1° grid, monthly | GFDL-ESM4, IPSL-CM6A-LR, MPI-ESM1-2-HR |
| Sea Ice Area Fraction | **siconc** | % | 1° grid, monthly | GFDL-ESM4, IPSL-CM6A-LR, MPI-ESM1-2-HR, UKESM1-0-LL |





| Sea water salinity | **so, so-bot, so-surf** | 0.001 | 1° grid, vertically resolved, ocean bottom and surface fields, monthly | GFDL-ESM4, IPSL-CM6A-LR, MPI-ESM1-2-HR, UKESM1-0-LL |
|---|---|---|---|---|
| Sea water potential temperature | **thetao** | °C | 1° grid, vertically resolved, monthly | GFDL-ESM4, IPSL-CM6A-LR, MPI-ESM1-2-HR, UKESM1-0-LL |
| Ocean model cell thickness | **thkcello** | m | 1° grid, vertically resolved, monthly | GFDL-ESM4, IPSL-CM6A-LR, MPI-ESM1-2-HR, UKESM1-0-LL |
| Sea water potential temperature at sea floor (bottom) | **tob** | °C | 1° grid, monthly | GFDL-ESM4, IPSL-CM6A-LR, MPI-ESM1-2-HR, UKESM1-0-LL |
| Sea surface temperature | **tos** | °C | 1° grid, monthly | GFDL-ESM4, IPSL-CM6A-LR, MPI-ESM1-2-HR, UKESM1-0-LL |
| Sea water zonal velocity | **uo** | m s-1 | 1° grid, vertically resolved, monthly | IPSL-CM6A-LR, MPI-ESM1-2-HR, UKESM1-0-LL |
| Sea water meridional velocity | **vo** | m s-1 | 1° grid, vertically resolved, monthly | IPSL-CM6A-LR, MPI-ESM1-2-HR, UKESM1-0-LL |
| Concentration of mesozooplankton expressed as carbon in sea water | **zmeso, zmeso-vint** | mol m-3 | 1° grid, vertically resolved and vertically integrated, monthly | GFDL-ESM4, IPSL-CM6A-LR, UKESM1-0-LL |
| Concentration of microzooplankton expressed as carbon in sea water | **zmicro, zmicro-vint** | mol m-3 | 1° grid, vertically resolved and vertically integrated, monthly | GFDL-ESM4, IPSL-CM6A-LR, UKESM1-0-LL |



| Total Zooplankton Carbon Concentration | **zooc, zooc-vint** | mol m-3 | 1° grid, vertically resolved and vertically integrated, monthly | GFDL-ESM4, IPSL-CM6A-LR, MPI-ESM1-2-HR, UKESM1-0-LL |
| --- | --- | --- | --- | --- |


### 2.4.2 Bias-adjusted global ocean forcings ('de-biased' sensitivity experiment)

GCMs have been shown to have limitations in accurately representing various aspects of the present
climate system (Eyring et al., 2023), (Séférian et al., 2020), that are also expected to affect regional
physical and biogeochemical oceanic projections (Li et al., 2016), (Tagliabue et al., 2021). In particular,
biases in sea-surface temperature (SST, variable 'tos') and nutricline as well as thermocline depth
influence oceanic primary productivity, which in turn has major influence on various marine ecosystem
processes. Thus, reducing the substantial biases in GCMs' ocean variables through bias-adjustment is
desirable. Typically, for bias-adjustment of atmospheric variables, statistical approaches are used where
a transfer function is trained to map the simulated historical distribution of the relevant variables to the
observed distribution and then applied to future simulations. Yet for oceanic variables, the scarcity of
comprehensive sub-surface observational data globally does not allow for a similar, direct adjustment of
the relevant variables. However, standalone ocean-biogeochemistry simulations, when driven by
observation-based atmospheric reanalysis data, have been demonstrated to considerably alleviate
SST-related biases and typically provide satisfactory simulations of the physical ocean and marine
biogeochemistry for the historical period (e.g. (Tsujino et al., 2020), (Barrier et al., 2023). Thus, an
alternative process-oriented bias-adjustment approach has been developed that relies on a
comprehensive ocean-biogeochemistry model that is forced by bias-adjusted atmospheric forcings. The
adjustment of the ISIMIP3b oceanic forcings builds on such a dynamical de-biasing approach (Lengaigne
et al., 2025), which relies on conducting forced oceanic simulations using the NEMO-PISCES
physical-biogeochemical ocean model (Madec, 2015), which is the oceanic component of the
IPSL-CM6A-LR climate model. The ocean model needs to be forced with high-frequency (3-hourly)
surface momentum, heat and freshwater fluxes. Since from the CMIP6 pre-industrial, historical, and
future scenario simulations used in ISIMIP3b these variables are only available at monthly resolution,
additional steps are necessary to generate climatological high-frequency fluxes first. In the following, we
first describe these preparatory steps, and then the de-biasing strategy, in more detail.

**High-frequency surface flux forcing.** Initially, a climatological simulation spanning the historical
period from 1958 to 2022 is performed by forcing the ocean model NEMO-PISCES with a single
repeating annual cycle representative of the 1990's climate conditions sourced from the "Repeat Year
Forcing" (RYF) from JRA55 reanalysis (Stewart et al., 2020). This simulation is driven using the CORE bulk
formulae (Large W. G., 2004), incorporating all surface atmospheric variables at 3-hourly resolution from
JRA55 RYF as inputs and storing 3-hourly momentum, heat and freshwater fluxes from this simulation.
These 3-hourly JRA55 RYF fluxes are the added to the monthly seasonal flux anomalies available from
the ISIMIP3b climate models for the pre-industrial (picontrol), historical (historical) and future SSP1-2.6
(ssp126), SSP3-7.0 (ssp370), and SSP5-8.5 (ssp585) scenarios. In this way, 3-hourly surface flux forcings



are created for all ISIMIP3b scenarios. Notably, this procedure results in sub-monthly variability
mirroring that of the JRA55 RYF, rather than the variability simulated by the coupled climate model. This
means that any projected changes in sub-monthly variability due to climate change are not integrated in
the final de-biased product. However, to date, marine ecosystem modellers have not analysed
sub-monthly variability anyways (and most marine ecosystem models are not suited to account for
sub-monthly variability of forcings), making this approach suitable.
Alternatively, de-biased ocean simulations including GCM-based sub-monthly variability could be
constructed by an alternative approach. In this scenario, 3-hourly surface atmospheric variables would
be extracted directly from each GCM simulation, rather than from JRA55 RYF forced oceanic simulations.
Forcing NEMO-PISCES with these variables using bulk formulae would once again produce the necessary
3-hourly surface fluxes, this time with variability consistent with the coupled GCM across all timescales.
This approach however requires running a separate ocean simulation for each GCM and scenario to
derive the surface fluxes, necessitating a much larger number of ocean model runs than the approach
using JRA55 RYF.  In addition, the 3-hourly input from the GCMs is not available without gaps.

**De-biasing strategy.** The 3-hourly surface fluxes, constructed as described above, then serve as forcings
for another set of ocean model simulations. Notably, these simulations are not driven with bulk
formulae but directly with surface fluxes to enable an online implementation of the surface heat flux
feedbacks triggered by climate change into the forced ocean biogeochemistry model for historical and
future simulations (Lengaigne et al., 2025). For bias-adjustment, the part of the anomalous surface
fluxes that directly depends on climate change-induced SST warming is separated from the part that
does not. Only the latter part is used as a direct flux input to the ocean model, while the former is
implemented within NEMO-PISCES as an online relaxation to the warming signal from the debiased
historical and future simulations using a spatially and seasonally variable feedback damping coefficient.
This SST feedback coefficient, derived from observed surface variables, represents the Newtonian
cooling negative feedback related to latent heat fluxes through the Clausius-Clapeyron relationship and
the negative feedback related to upward long-wave radiation through through Stefan's law (Zhang and Li
2014) and the positive downward longwave radiation feedback related to increasing temperature
(Shakespeare et al. 2022). Application of this approach to the ocean model effectively reproduces the
global SST changes simulated by CMIP6 models, as demonstrated in  (Lengaigne et al., 2025).
In this way, physical and biogeochemical ocean simulations are generated for picontrol and historical
climate forcings as well as for each of the future climate change scenarios, ensuring that the background
climatological state is constrained by the reanalysis, while still accounting for both interannual and
long-term climate variability simulated by the underlying GCM. Consequently, the resulting
ocean-biogeochemistry simulations considerably mitigate the strong present-day climatological biases
identified in the coupled models. Depending on data availability for the relevant monthly fluxes, this
de-biasing procedure can be applied to any climate model.
Additionally, to generate observation-based oceanic forcings for ISIMIP3a, a reference simulation is also
forced with the full JRA55 forcing (Tsujino et al. 2018) that includes observed inter-annual and decadal
variability. This oceanic forcing is expected to be a valuable additional climate-related forcing for impact
model evaluation within ISIMIP3a akin to the GFDL-MOM6-COBALT2 reanalysis-driven historical dataset





used in ISIMIP3a ((Frieler et al., 2024). The set of variables included in the de-biased dataset is a subset
to the one in the raw GCM dataset (Table 8), detailed  in Table 9.

**Table 9:** Bias-adjusted ocean data to be used by global impact models in the 'de-biased' sensitivity
experiment in the fisheries and marine ecosystems sector

| Variable | Variable specifier (variables in brackets are not directly available as model output but will have to be derived in post-processing) | Unit | Resolution | Forcing datasets |
|---|---|---|---|---|
| Mass concentration of total phytoplankton expressed as chlorophyll | **chl** | kg m-3 | 1° grid, vertically resolved, monthly | JRA55+IPSL-CM6A-LR |
| Sea floor depth | **deptho** | m | 1° grid, constant | JRA55+IPSL-CM6A-LR |
| Downward flux of particulate organic carbon | **expc-bot** | mol m-2 s-1 | 1° grid, monthly | JRA55+IPSL-CM6A-LR |
| Net primary organic carbon production by all types of phytoplankton | **intpp** | mol m-2 s-1 | 1° grid, monthly | JRA55+IPSL-CM6A-LR |
| Net primary organic carbon production by diatoms | **intppdiat** | mol m-2 s-1 | 1° grid, monthly | JRA55+IPSL-CM6A-LR |
| Net Primary Organic Carbon Production by Other Phytoplankton | **intppmisc** | mol m-2 s-1 | 1° grid, monthly | JRA55+IPSL-CM6A-LR |
| Mixed layer depth defined by delta rho = 0.125 | **mlotstmax** | m | 1° grid, monthly | JRA55+IPSL-CM6A-LR |
| Dissolved oxygen concentration | **o2, (o2-bot), o2-surf** | mol m-3 | 1° grid, vertically resolved, ocean bottom and surface fields, monthly | JRA55+IPSL-CM6A-LR |
| pH | **ph, (ph-bot), ph-surf** | 1 | 1° grid, vertically resolved, ocean | JRA55+IPSL-CM6A-LR |



| | | | bottom and surface fields, monthly | |
|---|---|---|---|---|
| Total phytoplankton carbon concentration | **phyc, (phyc-vint)** | mol m-3 | 1° grid, vertically resolved and vertically integrated, monthly | JRA55+IPSL-CM6A-LR |
| Concentration of diatoms expressed as carbon in sea water | **phydiat, (phydiat-vint)** | mol m-3 | 1° grid, vertically resolved and vertically integrated, monthly | JRA55+IPSL-CM6A-LR |
| Mole Content of Miscellaneous Phytoplankton Expressed as Carbon in Sea Water | **phymisc, (phymisc-vint)** | mol m-2 | 1° grid, vertically resolved and vertically integrated, monthly | JRA55+IPSL-CM6A-LR |
| Net Downward Shortwave Radiation at Sea Water Surface | **rsndts** | W m-2 | 1° grid, monthly | JRA55+IPSL-CM6A-LR |
| Sea water salinity | **so, (so-bot), so-surf** | 0.001 | 1° grid, vertically resolved, ocean bottom and surface fields, monthly | JRA55+IPSL-CM6A-LR |
| Sea water potential temperature | **thetao** | °C | 1° grid, vertically resolved, monthly | JRA55+IPSL-CM6A-LR |
| Ocean model cell thickness | **thkcello** | m | 1° grid, vertically resolved, monthly | JRA55+IPSL-CM6A-LR |
| Sea water potential temperature at sea floor (bottom) | **(tob)** | °C | 1° grid, monthly | JRA55+IPSL-CM6A-LR |
| Sea surface temperature | **tos** | °C | 1° grid, monthly | JRA55+IPSL-CM6A-LR |
| Sea water zonal velocity | **uo** | m s-1 | 1° grid, vertically resolved, monthly | JRA55+IPSL-CM6A-LR |
| Sea water meridional velocity | **vo** | m s-1 | 1° grid, vertically resolved, monthly | JRA55+IPSL-CM6A-LR |
| Concentration of mesozooplankton | **zmeso, (zmeso-vint)** | mol m-3 | 1° grid, vertically resolved and vertically | JRA55+IPSL-CM6A-LR |



| | | | | |
|---|---|---|---|---|
| expressed as carbon in sea water | | | integrated, monthly | |
| Concentration of microzooplankton expressed as carbon in sea water | **zmicro, (zmicro-vint)** | mol m-3 | 1° grid, vertically resolved and vertically integrated, monthly | JRA55+IPSL-CM6A-LR |
| Total Zooplankton Carbon Concentration | **zooc, (zooc-vint)** | mol m-3 | 1° grid, vertically resolved and vertically integrated, monthly | JRA55+IPSL-CM6A-LR |


### 2.4.3 Bias-adjusted regional ocean forcings ('de-biased' sensitivity experiment)

Regional marine ecosystem models are most often calibrated to reproduce observed environmental variables when driven by observed sea surface and bottom temperature, primary production (phytoplankton production), and zooplankton biomass. However, that would still lead to biases in the historical simulations if the impact model was forced by biased simulated input data instead of observational data. To reduce this effect the GCM-based input data has been adjusted such that the historical GCM simulations match observational data for certain regions (Eddy et al., 2025). The adjustment is based on the delta approach where simulated and observational forcing data $X_{sim}$ and $X_{obs}$ are averaged across a given historical reference period to determine the bias delta = mean ($X_{sim}$) - mean ($X_{obs}$) that is then subtracted from the simulated forcing data. This method preserves the trend in the forcing data and its internal variability. Some ocean forcing variables are not an exact match with variables used in regional marine ecosystem models. For example, sea water potential temperature (thetao), concentration of diatoms (phydiat-vint), or concentration of mesozooplankton (zmeso-vint) may first be converted to other indicators that are then used as input for the regional marine ecosystem models. In these cases the derived indicator is corrected using the delta method (see Table 10).

**Table 10:** Bias-adjusted ocean data to be used by regional impact models in the 'de-biased' sensitivity experiment in the fisheries and marine ecosystems sector

| Variable | Variable specifier | Unit | Resolution | Forcing datasets |
|---|---|---|---|---|
| Southern Benguela Current | | | | |
| Net primary organic carbon production by all types of phytoplankton | **intpp** | mol m-2 s-1 | 1° grid, monthly | Corrected based on observed primary production for the southern Benguela current based on the delta method where the adjustment target is data from 1978 for the EwE model and 1990 for the Atlantis model |



| Sea water potential temperature | **thetao** | °C | 1° grid, vertically resolved, monthly | Raw GCM temperature data converted to temperatures at 0-50, 50-100, 100-300 and 300-500 m according to the configuration for the southern Benguela Atlantis model, and 0-50 and 300-500 m for the EwE model. |
|---|---|---|---|---|
| **Cook Strait** | | | | |
| Net primary organic carbon production by all types of phytoplankton | **intpp** | mol m-2 s-1 | 1° grid, monthly | Corrected based on observed primary production for Cook Strait using the delta method where observational target data is from 1950 |
| **East Bass Strait** | | | | |
| Net primary organic carbon production by all types of phytoplankton | **intpp** | mol m-2 s-1 | 1° grid, monthly | Corrected based on observed primary production for East Bass Strait using the delta method where observational target data is from 1994 |
| **East Bering Sea** | | | | |
| Concentration of diatoms expressed as carbon in sea water | **phydiat-vint** | mol m-3 | 1/4° grid, vertically resolved and vertically integrated, monthly | Converted to phytoplankton size classes used in East Bering Sea mizer model then corrected using the delta method for the period 1982–1993 |
| Concentration of diazotrophs expressed as carbon in sea water | **phydiaz-vint** | mol m-3 | 1/4° grid, vertically resolved and vertically integrated, monthly | Converted to phytoplankton size classes used in East Bering Sea mizer model then corrected using the delta method for the period 1982–1993 |
| Concentration of picoplankton expressed as carbon in sea water | **phypico-vint** | mol m-3 | 1/4° grid, vertically resolved and vertically integrated, monthly | Converted to phytoplankton size classes used in East Bering Sea mizer model then corrected using the delta method for the period 1982–1993 |
| Concentration of mesozooplankton expressed as carbon in sea water | **zmeso-vint** | mol m-3 | 1/4° grid, vertically resolved and vertically integrated, monthly | Converted to zooplankton size classes used in East Bering Sea mizer model then corrected using the delta method for the period 1982–1993 |



| | | | | |
|---|---|---|---|---|
| Concentration of microzooplankton expressed as carbon in sea water | **zmicro-vint** | mol m-3 | 1/4° grid, vertically resolved and vertically integrated, monthly | Converted to zooplankton size classes used in East Bering Sea mizer model then corrected using the delta method for the period 1982-1993 |
| Sea surface temperature | **tos** | °C | 1/4° grid, monthly | Corrected based on configuration for the East Bering Sea mizer model using the delta method for the period 1982–1993 |
| Hawai'i | | | | |
| Concentration of diatoms expressed as carbon in sea water | **phydiat-vint** | mol m-3 | 1/4° grid, vertically resolved and vertically integrated, monthly | Converted to phytoplankton size classes used in Hawaii mizer model (Woodworth-Jefcoats, 2022) then corrected using the delta method |
| Concentration of diazotrophs expressed as carbon in sea water | **phydiaz-vint** | mol m-3 | 1/4° grid, vertically resolved and vertically integrated, monthly | Converted to phytoplankton size classes used in Hawaii mizer model then corrected using the delta method |
| Concentration of picoplankton expressed as carbon in sea water | **phypico-vint** | mol m-3 | 1/4° grid, vertically resolved and vertically integrated, monthly | Converted to phytoplankton size classes used in Hawaii mizer model then corrected using the delta method |
| Concentration of mesozooplankton expressed as carbon in sea water | **zmeso-vint** | mol m-3 | 1/4° grid, vertically resolved and vertically integrated, monthly | Converted to zooplankton size classes used in Hawaii mizer model then corrected using the delta method |
| Concentration of microzooplankton expressed as carbon in sea water | **zmicro-vint** | mol m-3 | 1/4° grid, vertically resolved and vertically integrated, monthly | Converted to zooplankton size classes used in Hawaii mizer model then corrected using the delta method |
| Sea water potential temperature | **thetao** | °C | 1/4° grid, vertically resolved, monthly | Converted to temperature used in Hawaii Mizer model then corrected based on observed sea water potential temperature for Hawaii using the delta method from |



| | | | | 1961–1980 with observed temperature data from the World Ocean Atlas |
|---|---|---|---|---|



## 2.5 Future Lightning Data

For the 'varlighting' sensitivity experiment we provide temporally varying lightning density (strokes km$^{-2}$
day$^{-1}$) for the period 2015-2100 on monthly resolution (monthly mean of daily lightning stroke density)
and the standard 0.5° global grid. This dataset may be used in a range of applications, for example, to
understand the influence of lightning on wildfire ignition or atmospheric composition.
The lightning density is derived from future climate simulations by UKESM1-0-LL and an empirical
relationship between Convective Available Potential Energy (CAPE) and lightning strokes based on the
WWLLN Global Lightning Climatology and time-series (WGLC) (Kaplan & Lau, 2021, 2022). Daily mean
CAPE is calculated from non bias-adjusted air temperature, air pressure, and specific humidity on
pressure levels from the surface to the top of the troposphere.
The relationship between daily CAPE and daily lightning is estimated by linear regression of
log-transformed CAPE derived from the GCM-calculated CAPE during the period of overlapping model
output and observed daily lightning from WGLC (2015-2020) for each gridcell and month of the year.
Where <10 observations of daily lightning were available over the calibration period, we used global
mean regression parameters.
The empirical relationships are applied to the daily CAPE data from the UKESM1-0-LL simulations for all
three climate scenarios SSP1-2.6, SSP3-7.0, and SSP5-8.5. The associated lightning densities were
monthly averaged. To maintain the spatial structure of lightning observed at present, lightning
anomalies compared to the simulated 2015-2020 climatological reference were added to the observed
present-day lightning climatology from WGLC for 2015-2020. The 'varlighning' sensitivity experiment is
assumed to start from the default historical group I simulation, assuming the Flash Rate Monthly
Climatology (Cecil, 2006), not changing with climate change.


**Table 11:** Future lightning forcing data provided within ISIMIP3b.

| Variable | Variable specifier | Unit | Resolution | Datasets |
|---|---|---|---|---|
| Monthly flash rate | **lightning** | km-2 d-1 | 0.5° grid, monthly | Derived from UKESM1-0-LL (SSP1-2.6, SSP3-7.0, and SSP5-8.5) using an empirical relationship between Convective Available |





| | | | | Potential Energy (CAPE) and lightning densities (Kaplan et al., 2023). |
|---|---|---|---|---|

**3 Conclusions**

This paper gives an overview over the ISIMIP3b, group I and II experiments and the provided climate-related forcing data sets. The simulations assuming fixed 2015 direct human forcings and a low (ssp126) and two high emission scenarios (ssp370 and ssp585) are designed to describe the impacts of different levels of climate change on present day natural and human systems. The set-up allows e.g. for testing to what degree the (bio-)physical impacts scale with global mean temperature change and could therefore be translated to other global warming pathways than the ones considered here. While a functional relationship between the considered impact indicator and global mean temperature change (or other climate variables) could be trained on ssp585 simulations because of the high warming levels reached, its performance could then be tested on ssp370 and ssp126. However, in such a setting it has to be taken into account that ssp370 is different from the other scenarios with regard to particularly high aerosol emissions and high decreases in forest areas going beyond the assumptions in the other models. So it has been shown that the increase of global mean precipitation with global warming is much weaker in SSP3-7.0 than in the other scenarios (Shiogama et al., 2023).

This paper is intended to work as a catalogue where the climate impact modellers can find all relevant information about the climate-related forcings needed as reference for the impact model simulations generated within the CMIP6-based ISIMIP3b, group I (historical period) and group II (future projections). As a continuous process we would like to improve or complement these data sets wherever possible. So this paper can also be read as a call to either contribute by additional input data that allows other sectors to join the current simulation round or by methods that could be used to generate additional data sets for the next simulation round that will likely build on CMIP7 simulations. The following climate-related forcings have been identified as still missing and particularly critical to be added to a fourth simulation round of ISIMIP: i) temporally resolved lightning data accounting for changes in climate, ii) bias-adjusted oceanic forcing data, iii) projected coastal water levels in high temporal resolution accounting for extremes and representing the effects of long term sea level rise in line with the underlying global climate simulations, and v) ozone concentration fields in line with the GCM simulations. While a bias-adjustment of the oceanic forcings is already suggested in section **2.4.2,** the approach does not preserve the daily variability of the raw oceanic forcings as it requires atmospheric surface flux only available in monthly resolution from the ISIMIP3b GCMs. To ensure the consistency on daily time scale, we have submitted an associated request for CMIP7 whose simulations will be used within the next round of ISIMIP. The generation of high resolution coastal water levels is ongoing research described in section **2.2.3.** In particular the generation of the short term variability that will have to be added to the long term trends in water levels still has to be developed and prove to fulfill the demands. In addition, it would be great to also provide estimates of the extreme coastal water levels associated with the tropical cyclone tracks and wind fields we provide within ISIMIP3b (see section **2.2**).





There is a general demand for higher resolution climate-related forcings including both, the oceanic and the atmospheric components ideally accounting for heat island effects. As the ISIMIP climate-related forcings have to be globally consistent in the sense that they have to represent the daily variability of the underlying coarse resolution GCMs, we cannot use data from dynamical downscaling approaches using boundary conditions from different GCM runs as for example available through CORDEX. However, it seems to be appealing to harmonize the selection of the ISIMIP GCMs with a priority setting regarding the GCM-based boundary conditions within CORDEX.

The climate-related forcings described here are also provided as input for the new ISIMIP3b, group III simulations where the associated Direct Human Forcings (DHF) are not held constant at 2015 levels but are projected into the future in line with i) the population growth and economic development associated with the considered Shared Socioeconomic Pathways (SSPs) and mitigation measures required to reach the prescribed levels of climate forcings associated with the climate projections ('no adaptation' experiments) and ii) additionally accounting for the impacts of climate change ('adaptation' experiments). The collection of the associated DHF will be described in a separate paper.

**Code and data availability.** The MIT data on tropical cyclone tracks with wind and precipitation fields data shall be used for non-commercial research or academic purposes only. Data can be made available by the ISIMIP team upon written consent by Kerry Emanuel (MIT, email: emanuel@mit.edu).
All other input data described are available for participating modelers with a respective account from the DKRZ server. Data will be made publicly available, and most data are already publicly available at the ISIMIP data repository at https://data.isimip.org/ (ISIMIP data repository, 2025) and availability is documented in the ISIMIP Input data table https://www.isimip.org/gettingstarted/input-data-bias-adjustment/ (ISIMIP input data table, 2025) where the way to access the data is described as well. Model output is already partly available https://data.isimip.org/ (ISIMIP data repository, 2025). The ISIMIP repository fulfills the archive standards as stated in the "GMD code and data policy". The repository is hosted and maintained by the Potsdam Institute for Climate Impact Research (PIK). Data can only be published or removed from the repository by the ISIMIP data team, which is monitored by the ISIMIP steering committee according to the organizational structure of ISIMIP. DOIs are used to refer to datasets in a persistent way. Whenever a dataset is replaced for any reason a copy is kept on tape, and a new DOI is issued, while the old DOI is kept online with information on how to retrieve the archived data. Detailed information can be found in the ISIMIP terms of use at https://www.isimip.org/gettingstarted/terms-of-use/ (ISIMIP terms of use, 2023).

**Author contributions**
KF lead the project and developed the concept with contributions from JS, MM, CO, CPOR, SH, JLB, CSH, CMP, TDE, KOC, CN, RH, DPT, OM, SJC, JJ, SR, GL, SC, EB, AGS, NS, JC, SH, CB, AG, FL, SNG, HMS, FH, TH, RM, DP, WT, DMB, RL, AIA, MF, MB, RR, and IDG. JV, MB, JK, IDVDV, LN, IJS supported the quality control and curation of the climate-related forcing data and the protocol development together with the sectoral ISIMIP coordinators listed as co-authors. SL developed the method and generated the downscaled and bias-adjusted atmospheric climate forcing data. MM and ST provided the description of



the approach to generate the coastal water level data. ML provided the description of the method to bias-adjust the global oceanic forcings. TV, DQC, CYL, SJC, and KE provided TC data. JOK and AK provided the future lightning data. KF prepared the manuscript with contributions from all co-authors.

**Competing interests**

At least one of the (co-)authors is a member of the editorial board of GMD

**Acknowledgements**

This article is based upon work from COST Action CA19139 PROCLIAS (PROcess-based models for CLimate Impact Attribution across Sectors), supported by COST (European Cooperation in Science and Technology; https://www.cost.eu). SL received funding from the German Research Foundation (DFG, project number 427397136). MB acknowledges funding from the BELSPO STEREO IV project SR/00/414. SC, AGS, MB and NS acknowledge funding through NERC NE/V01854X/1 (MOTHERSHIP). This research has received funding from the German Federal Ministry of Education and Research (BMBF) under the research projects QUIDIC (grant agreement no. 01LP1907A) and ISI-Access (16QK05), from the Horizon 2020 Framework Programme of the European Union under the projects RECEIPT (grant agreement no. 820712). C.-Y. L is supported by Palisades Geophysical Institute (PGI) Young Scientist Award. KOC acknowledges support from the National Research Foundation of South Africa (grant 136481) and the resources from the Cluster for High Performance Computing-CSIR. FL is supported by the National Key Research and Development Program of China (2022YFE0106500). DPT acknowledges funding from the Jarislowsky Foundation and NSERC. IDG acknowledges funding of the European Research Council (ERC Starting Grant, GROW-101041110).



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
