# Peer review of "Scenario set-up and the new CMIP6-based climate-related forcings provided within the"

_EGUsphere, 2025_

## Referee Comment (RC2)

**Review of egusphere-2025-2103**

*Scenario set-up and the new CMIP6-based climate-related forcings provided within the third round of the Inter-Sectoral Model Intercomparison Project (ISIMIP3b, group I and II)*

Anonymous Reviewer

**Summary**

The paper provides a general description of the climate-relative forcings, based on climate simulations performed during CMIP6, and improvements made for realistic representation of various climate forcings in the third simulation round of the Inter-Sectoral Impact Model Intercomparison Project (ISIMIP3b). The paper serves as a reference to perform impact model simulations using the forcing specifications described here, which can be used to evaluate the impact of climate forcings in different scenarios.

This is not a particularly long paper since most of it consists of table sand lists, but it is riddled with references pointing to incorrect tables and especially non-existent ones, whose contents cannot be found anywhere on the manuscript. As such, I was not able to comprehensively review this manuscript and recommend that the manuscript go through another round of revision.

**Comments**

**L93**

I know that the description has been given in the first paper for different sectors being considered; however, I'd like to see a concise summary of these sectors in the introduction, especially because too many references have been made here about how this work influences different sectors, but it is difficult to see what they are exactly.

**L144**

This sentence needs some proof-reading. The scenarios were chosen to "capture a wide range of possible futures", then "the availability of climate model simulations"? Perhaps the authors meant to "utilize" the availability or something along those lines?

**L155**

"Only" sounds odd here. Perhaps the authors meant "eventually"?

**L157**

Wrong quotation mark.

**L158**

I have two issues with this whole discussion on the *plausibility* of different scenarios. The first is that it has been claimed that the scenarios were chosen to "capture a wide range of possible futures" (L144), but the only thing that is discussed here is how some of the SSP scenarios are unrealistic. I do agree that some SSP scenarios are unlikely [1], but I do not see anything particularly *wide* regarding the choices. I do not think all future predictions must include extreme cases, but I am just not fully convinced of the stated goal of the ISIMIP3b scenarios. The second is that it is not clear

what the authors consider as the "business as usual" scenario. What do the authors actually think of the "business as usual" scenario, now that they have refuted almost all scenarios used in ISIMIP3b? My main concern is that the authors leave a number of open questions regarding the choice of plausible scenarios.

**L205**

To be honest, I cannot keep track of all the errors when it comes to references to figures and tables in this paper. Here, "Table 11" should be the correct one. I have also found cases where the authors refer to tables that simply do not exist. The manuscript needs an overhaul.

**L228**

This should be "to include".

**L229**

I believe it should be "impact distribution".

**L237**

I believe the authors are referring to "Table 2". I will not comment further about this, but always directly refer to the table being discussed, not "the table".

**L298**

I do not think a comma is needed here.

**L309**

"Too sparse"?

**L333**

Again, wrong table.

**L453**

I think it is unusual to only write the variable names. The authors did write "sea level pressure (psl)" at one point, and I think this should be the convention. Write both the variables names and the actual names of the properties being referenced so the readers will not have to go back to the table every time they encounter a variable name, at least for the first time.

**L479**

Table 16 does not exist.

**L512**

I do not find it particularly significant that the mean ECS is exactly the same as that of CMIP6, as it is a subset of CMIP6 ensemble. However, one important aspect of the estimation of ECS in CMIP6 is

that the variability in the estimates of ECS is still very large [2], [3]. I would like to see a detailed discussion on the variability of ECS, especially on how various factors influence the estimate of ECS in each GCM.

For example, the five GCMs chosen for this study seem to produce short-wave and long-wave radiative effects from clouds that are consistently close to median RMSD of the CMIP6 ensemble, and I would assume that as a result, the variability should be much smaller than the whole CMIP6 ensemble. Is it really the case? If not, what is driving the variability?

**L518**

Table 12 does not exist. At this point, there are too many errors involved in references to its own tables. I will not mention them from now on, but I do find it very odd that there are a number of references to tables that simply do not exist in the manuscript.

**L548**

What do you mean that the new approach "improves the spatial variability"?

**L577**

From this line on, references appear with parentheses where they should not.

**L612**

Why was the 5% threshold chosen? What was the reasoning behind this specific number?

**L679**

"Artifacts"

**L722**

I am not sure what I am supposed to find out in Table 4, which does not include any specifications of forcings or GCM data. This could be another case of incorrect table reference, but I cannot find the relevant information anywhere on the submitted manuscript.

**L783**

I am fairly certain that this is against GMD policy.

**L787**

What is the point of having the "Datasets" column if they are all the same? Surely you could have just added a note just for the surface temperature, or write it down in caption?

**Bibliography**

[1]  R. Pielke Jr, M. G. Burgess, and J. Ritchie, "Plausible 2005–2050 emissions scenarios project between 2 °C and 3 °C of warming by 2100," *Environmental Research Letters*, vol. 17, no. 2, p. 24027, Feb. 2022, doi: 10.1088/1748-9326/ac4ebf.

[2]  N. Scafetta, "Advanced Testing of Low, Medium, and High ECS CMIP6 GCM Simulations Versus ERA5-T2m," *Geophysical Research Letters*, vol. 49, no. 6, Mar. 2022, doi: 10.1029/2022gl097716.

[3]  M. D. Zelinka *et al.*, "Causes of Higher Climate Sensitivity in CMIP6 Models," *Geophysical Research Letters*, vol. 47, no. 1, Jan. 2020, doi: 10.1029/2019gl085782.

---

## Author Comment (AC2)

**ISIMIP3b paper review comments**

**Reviewer 2:**

**Summary**

The paper provides a general description of the climate-relative forcings, based on climate simulations performed during CMIP6, and improvements made for realistic representation of various climate forcings in the third simulation round of the Inter-Sectoral Impact Model Intercomparison Project (ISIMIP3b). The paper serves as a reference to perform impact model simulations using the forcing specifications described here, which can be used to evaluate the impact of climate forcings in different scenarios.

This is not a particularly long paper since most of it consists of tables and lists, but it is riddled with references pointing to incorrect tables and especially non-existent ones, whose contents cannot be found anywhere on the manuscript. As such, I was not able to comprehensively review this manuscript and recommend that the manuscript go through another round of revision.

**Reply:** Thanks so much for the careful review. There were indeed some flaws regarding table references and we apologize for that. We corrected these and are happy to receive further comments.

**Comments**

**L93:** I know that the description has been given in the first paper for different sectors being considered; however, I'd like to see a concise summary of these sectors in the introduction, especially because too many references have been made here about how this work influences different sectors, but it is difficult to see what they are exactly.

**Reply:** Thanks for pointing this out. We added the following to the respective sentence in the introduction: "... provides a common scenario framework for cross-sectorally consistent climate impact simulations. Currently, operational simulation protocols exist for the following sectors: Agriculture, Biomes, Energy, Fire, Food security and nutrition, Groundwater, Labour, Lakes global, Lakes regional, Fisheries and marine ecosystems global, Fisheries and marine ecosystems local, Peatland, Permafrost, Water global, Water regional. Additional protocols for  Coastal systems, Regional forests, Temperature-related mortality, health indicators, Terrestrial biodiversity and Water quality sectors are under development."

**L144:** This sentence needs some proof-reading. The scenarios were chosen to "capture a wide range of possible futures", then "the availability of climate model simulations"? Perhaps the authors meant to "utilize" the availability or something along those lines?

**Reply:** Thank you very much for the hint. The sentence has been changed to: "The selection of ISIMIP3b scenarios (see **Figure 1**) was generally driven by i)  the aim to capture a wide range of possible futures from low to high emission scenarios and,  provide of a long baseline simulation

assuming pre-industrial climate conditions that allows for a robust estimation of reference return levels of extreme events and ii) the availability of climate model simulations."

**L155:** "Only" sounds odd here. Perhaps the authors meant "eventually"?

**Reply:** Yes, the text has been adjusted accordingly.

**L157:** Wrong quotation mark.

**Reply:** Thank you very much! The text has been changed accordingly.

**L158:** I have two issues with this whole discussion on the plausibility of different scenarios. The first is that it has been claimed that the scenarios were chosen to "capture a wide range of possible futures" (L144), but the only thing that is discussed here is how some of the SSP scenarios are unrealistic. I do agree that some SSP scenarios are unlikely [1], but I do not see anything particularly wide regarding the choices. I do not think all future predictions must include extreme cases, but I am just not fully convinced of the stated goal of the ISIMIP3b scenarios. The second is that it is not clear what the authors consider as the "business as usual" scenario. What do the authors actually think of the "business as usual" scenario, now that they have refuted almost all scenarios used in ISIMIP3b? My main concern is that the authors leave a number of open questions regarding the choice of plausible scenarios.

**Reply:** The selection of the scenarios is a community-driven process constrained by the availability of climate model simulations (multi-GCM ensemble per scenario) and socio-economic background information (such as land use patterns, populations and GDP data etc. additionally required as 'Direct Human Forcing' for the ISIMIP3b, group III impact model simulations that will be introduced in an upcoming paper). These criteria have made CMIP6-ScenarioMIP our reference point for the selection. The 'Tier 1' ScenarioMIP scenarios comprise only four scenarios: SSP5-8.5, SSP3-7.0, SSP2-4.5, and SSP1-2.6 where 'Tier 1 spans a wide range of uncertainty in future forcing pathways important for research in climate science, IAM, and IAV studies, while also providing key scenarios to anchor experiments in a number of other MIPs (see last column in Table 2)' (O'Neill et al. 2016)). When saying that we aim 'to capture a wide range of possible futures' we refer to selecting the highest and the lowest scenario from the CMIP6-ScenarioMIP ensemble. SSP3-7.0 was added in a second step after a discussion among the sectoral coordinators and at the ISIMIP workshops in response to e.g., (Hausfather and Peters 2020). However, it is important to note, that none of the four ScenarioMIP scenarios is considered a 'business as usual scenario' and within ISIMIP do not do so and have never done so either. The critique by Hausfather et al., does not refer to the SSPs but the high emissions of RCP8.5. We modified the paragraph to highlight the clear linkage to CMIP6-ScenarioMIP and to underline that there simply is no 'business as usual scenario' and that there was not even the intention to design one within ScenarioMIP. The paragraph now reads:

'The selection of the scenarios is a community-driven process constrained by the availability of climate model simulations (multi-GCM ensemble per scenario) and socio-economic background information (such as land use patterns, populations and GDP data etc. additionally required as 'Direct Human Forcing' for the ISIMIP3b, group III impact model simulations that will be introduced in an

upcoming paper). These criteria have made CMIP6-ScenarioMIP the reference point for the selection (O'Neill et al. 2016). The selection of ISIMIP3b scenarios (see **Figure 1**) from the four ScenarioMIP Tier 1 scenarios was additionally driven by the aim to capture a wide range of possible futures from low to high emission scenarios and to provide of a long baseline simulation assuming pre-industrial climate conditions that allows for a robust estimation of reference return levels of extreme events. This is why the original selection comprised the pre-industrial baseline ('picontrol'), the historical simulations ('historical'), SSP1-2.6 representing the 'low end of the range of future forcing pathways in the IAM literature' (O'Neill et al. 2016), and SSP5-8.5 representing the 'high end of the range of future pathways in the IAM literature' (O'Neill et al. 2016). Given recent mitigation efforts, some estimates of recoverable coal reserves, and decreasing prices for renewable energies the emissions underlying SSP5-8.5 have been criticised for being unplausibly high (Hausfather and Peters 2020). Based on these discussions, the 'medium to high end of the range of future forcing pathway' SSP3-7.0 (O'Neill et al. 2016) has been added to the ISIMIP3b scenario set-up. While this scenario is described as 'average no climate protection policy' by (Hausfather and Peters 2020), we highlight that we explicitly do not consider it as a 'business as usual scenario' and that this was not the framing within ScenarioMIP either. Instead SSP3-7.0 is based on rather extreme assumptions about land use changes and aerosol emissions e.g. leading to a scaling of precipitation with global mean temperature that diverges from the scaling identified in the other scenarios (Shiogama et al. 2023). In addition, SSP5-8.5 is explicitly kept in the ISIMIP3b ensemble as its particularly strong warming signal allows testing to what degree the simulated impacts of climate may scale with global mean temperature, which could allow for a translation of impacts to other emission scenarios. In addition, even under lower emission scenarios, global warming levels as the ones reached under SSP5-8.5 in 2100 will eventually be reached later in time as long as emissions are not reduced to zero. These impacts of high warming levels would not be captured when only considering lower emission scenarios ending in 2100.'

**L205:** To be honest, I cannot keep track of all the errors when it comes to references to figures and tables in this paper. Here, "Table 11" should be the correct one. I have also found cases where the authors refer to tables that simply do not exist. The manuscript needs an overhaul.

**Reply:** Thanks, we reviewed all table and figure references and corrected all errors.

**L228:** This should be "to include".

**Reply:** True, thanks.

**L229:** I believe it should be "impact distribution".

**Reply:** We think of extreme value distribution as a certain type of statistical distribution. To clarify that we have modified the sentence in the following way:

'In order to allow for the fitting of extreme value distributions such as Gumble or Generalized Extreme Value (GEV) distributions to e.g., annual maximum river discharge to estimate reference 100 year return levels….'

**L237:** I believe the authors are referring to "Table 2". I will not comment further about this, but always directly refer to the table being discussed, not "the table".

**Reply:** Correct, thanks.

**L298:** I do not think a comma is needed here.

**Reply:** Corrected, thanks.

**L309:** "Too sparse"?

**Reply:** Indeed, thanks.

**L333:** Again, wrong table.

**Reply:** Changed to Table 2.

**L453:** I think it is unusual to only write the variable names. The authors did write "sea level pressure (psl)" at one point, and I think this should be the convention. Write both the variables names and the actual names of the properties being referenced so the readers will not have to go back to the table every time they encounter a variable name, at least for the first time.

**Reply:** Done

**L479:** Table 16 does not exist.

**Reply:** Thanks, should be Table 8.

**L512:** I do not find it particularly significant that the mean ECS is exactly the same as that of CMIP6, as it is a subset of CMIP6 ensemble. However, one important aspect of the estimation of ECS in CMIP6 is that the variability in the estimates of ECS is still very large [2], [3]. I would like to see a detailed discussion on the variability of ECS, especially on how various factors influence the estimate of ECS in each GCM. For example, the five GCMs chosen for this study seem to produce short-wave and long-wave radiative effects from clouds that are consistently close to median RMSD of the CMIP6 ensemble, and I would assume that as a result, the variability should be much smaller than the whole CMIP6 ensemble. Is it really the case? If not, what is driving the variability?

**Reply:** It's true that the RMSD in these two variables is relatively similar across our five models, and somewhat lower than in the CMIP6 multi-model median (Fig. 2). This is no coincidence, since the relative RMSD shown in Fig. 2 was used as one of several criteria for selecting models for ISIMIP3b; as explained higher up in the same section. The five selected GCMs therefore "by construction" tend to have relatively low RMSD across the set of variables tested in Fig. 2.

However, Fig. 2 shows the (dis-)agreement of models with historical observations, in terms of spatiotemporal variability in different variables; it does not show the variability itself. Therefore, the fact that the five selected models have low RMSD compared to the CMIP6 median does not imply that they exhibit low variability, in terms of cloud radiative effect. Moreover, even though

cloud-related processes appear to play an important role in explaining differences in ECS across models, these relationships are complex and differences cannot be easily attributed to individual processes (Zelinka et al. 2020; Meehl et al. 2020). Most importantly, the two variables in question - lwcre and swcre - measure the *short-term* cloud radiative effect, which has no correlation with the *long-term* cloud feedback that defines ECS (Chao, Zelinka, and Dessler 2024). Thus, the results shown in Fig. 2 are useful for assessing model skill compared to historical observations, but do not allow for any conclusions about the spread in ECS between different models. Generally, different models could achieve a similar agreement with observations through various different combinations of processes and feedback strengths; which is one reason why model estimates of ECS still diverge so strongly. The five models selected here reflect this divergence in estimates, while all being in relatively good agreement with historical observations, which in our opinion makes them a reasonable choice for studying climate impacts (noting that data availability additionally constrained model selection).

We have expanded our discussion of ECS in the climate models, in response to this comment as well as a related comment by reviewer 1, as follows:

"The five GCMs provide a good representation of both the mean and the range of the full CMIP6 multi-model ensemble ECS. According to (Meehl et al. 2020), the CMIP6 multi-model mean ECS is 3.7°C, which is precisely met by the mean ECS of the five ISIMIP3b GCMs. The transient climate response (TCR) of 2.0°C is also precisely met. This provides an improvement over ISIMIP2b, in the sense of the selected GCM subset reflecting the statistics of the larger CMIP ensemble. In ISIMIP2b the mean ECS for the full CMIP5 was 3.2°C compared with a mean ECS of 3.72°C for the four ISMIP2b GCMs (see Table S1 and S2 in (Jägermeyr et al. 2021)). The ISIMIP3b ensemble includes three models with below-average ECS (GFDL-ESM4, MPI-ESM1-2-HR, MRI-ESM2-0) and two models with above-average ECS (IPSL-CM6A-LR, UKESM1-0-LL) (see **Table 4**). In line with their ECS values, we find GFDL-ESM4 and UKESM1-0-LL to project the weakest and strongest global warming, respectively, under any future scenario considered (see **Figure 3**). Under SSP5-8.5, the global mean near-surface temperature in 2100 is about 3°C larger in UKESM1-0-LL than in GFDL-ESM4. Under SSP1-2.6, the projections are about 1.5°C apart. The ensemble mean warming of the ISIMIP3b CMIP6 models is significantly higher than the warming of the ISIMIP2b CMIP5 models, across global land area by an average of 0.3°C, but over the main breadbasket cropland regions by more than 0.5°C between 1983–2013 and 2069–2099, under both SSP1-2.6 and SSP5-8.5 (Table S1 in (Jägermeyr et al. 2021). This is in line with the higher median ECS in CMIP6 compared to CMIP5; indeed, some CMIP6 models have an ECS above the assessed likely (2.5°C to 4°C) and very likely (2°C to 5°C) ranges in the IPCC's sixth assessment report (AR6) (Forster et al. 2021). The reasons for these higher estimates of ECS are complex, with cloud feedback processes playing an important role (Zelinka et al. 2020). While the plausibility of the very high ECS estimates has been questioned, recent studies indicate CMIP6 models with high ECS tend to simulate cloud properties better than low ECS models (Bock and Lauer 2024); also, unaccounted natural variability may have biased the IPCC's assessed ranges somewhat low (Watanabe et al. 2024; Liang, Gillett, and Monahan 2024).

The ISIMIP3b ensemble reflects the spread in ECS of the overall CMIP6 ensemble, with two models above the AR6 likely range and one of these (UKESM1-0-LL) above the very likely range. The strong warming response of these models should be kept in mind when conducting ISIMIP3b-based impacts studies. However, depending on the region and variable of interest, the high ECS does not necessarily

have any bearing on the magnitude or realism of projected regional impacts, and any further selection of models should not be based solely on ECS but on the models' suitability for the impacts variables in question (Swaminathan et al. 2024). In many applications, results can be harmonized by describing the simulated impacts in terms of global mean temperature changes instead of time for the different emission scenarios."

**L518:** Table 12 does not exist. At this point, there are too many errors involved in references to its own tables. I will not mention them from now on, but I do find it very odd that there are a number of references to tables that simply do not exist in the manuscript.

**Reply:** Yes, we wanted to point to Table 4 "Characteristics of CMIP6 climate models used in ISIMIP3b". The text has been modified accordingly.

**L548:** What do you mean that the new approach "improves the spatial variability"?

**Reply:** A simple interpolation from coarser resolution data to high resolution underestimates the spatial variability within the grid cells. In the new approach this is avoided by adding additional variability at the target resolution. We have modified the sentence to:

"Compared to ISIMIP2b, where climate model output was first spatially interpolated to the target resolution and then bias-adjusted, the new approach avoids the associated underestimation of the spatial variability at the target resolution (Lange 2019)."

**L577:** From this line on, references appear with parentheses where they should not.

**Reply:** Has been adjusted.

**L612:** Why was the 5% threshold chosen? What was the reasoning behind this specific number?

**Reply:** We used the 5% significance threshold because it is a widely accepted convention in statistics, corresponding to a 95% confidence level and roughly ±2 standard deviations (±1.96) under the normal distribution. This level is commonly applied in many areas of research as it balances the risk of false positives with maintaining sufficient statistical power, whereas stricter (1%) or more lenient (10%) thresholds are typically reserved for specific contexts.

**L679:** "Artifacts"

**Reply:** Thanks, corrected.

**L722:** I am not sure what I am supposed to find out in Table 4, which does not include any specifications of forcings or GCM data. This could be another case of incorrect table reference, but I cannot find the relevant information anywhere on the submitted manuscript.

**Reply:** We wanted to point to Table 6 where the GCM data used to force the TC models are listed. The text has been adjusted accordingly.

**L783:** I am fairly certain that this is against GMD policy.

**Reply:** We have revised the Data and code availability section and implemented an automatic process into the ISIMIP repository to retrieve the data upon signing a non-commercial license. We addressed the issue in detail in the answer to the related chief editors comment.

**L787:** What is the point of having the "Datasets" column if they are all the same? Surely you could have just added a note just for the surface temperature, or write it down in caption?

**Reply:** Besides others, the paper serves as a look-up document for modelers performing climate impact simulations in line with the ISIMIP protocol. To make information as easy to find as possible and to avoid confusion we decided to stick to a common and systematic table layout also in cases where this may result in very short tables, or repeated table entries.

Bibliography

Bock, Lisa, and Axel Lauer. 2024. "Cloud Properties and Their Projected Changes in CMIP Models with Low to High Climate Sensitivity." *Atmospheric Chemistry and Physics* 24 (3): 1587–1605.

Chao, Li-Wei, Mark D. Zelinka, and Andrew E. Dessler. 2024. "Evaluating Cloud Feedback Components in Observations and Their Representation in Climate Models." *Journal of Geophysical Research Atmospheres* 129 (2). https://doi.org/10.1029/2023jd039427.

Forster, P., T. Storelvmo, K. Armour, W. Collins, J-L Dufresne, D. Frame, D. Lunt, et al. 2021. *Chapter 7: The Earth's Energy Budget, Climate Feedbacks, and Climate Sensitivity. Climate Change 2021: The Physical Science Basis*.

Hausfather, Zeke, and Glen P. Peters. 2020. "Emissions – the 'business as Usual' Story Is Misleading." Nature Publishing Group UK. January 29, 2020. https://doi.org/10.1038/d41586-020-00177-3.

Jägermeyr, Jonas, Christoph Müller, Alex C. Ruane, Joshua Elliott, Juraj Balkovic, Oscar Castillo, Babacar Faye, et al. 2021. "Climate Impacts on Global Agriculture Emerge Earlier in New Generation of Climate and Crop Models." *Nature Food* 2 (11): 873–85.

Lange, Stefan. 2019. "Trend-Preserving Bias Adjustment and Statistical Downscaling with ISIMIP3BASD (v1.0)." *Geoscientific Model Development* 12 (7): 3055–70.

Liang, Yongxiao, Nathan P. Gillett, and Adam H. Monahan. 2024. "Accounting for Pacific Climate Variability Increases Projected Global Warming." *Nature Climate Change* 14 (6): 608–14.

Meehl, Gerald A., Catherine A. Senior, Veronika Eyring, Gregory Flato, Jean-Francois Lamarque, Ronald J. Stouffer, Karl E. Taylor, and Manuel Schlund. 2020. "Context for Interpreting Equilibrium Climate Sensitivity and Transient Climate Response from the CMIP6 Earth System Models." *Science Advances* 6 (26): eaba1981.

O'Neill, Brian C., Claudia Tebaldi, Detlef P. van Vuuren, Veronika Eyring, Pierre Friedlingstein, George Hurtt, Reto Knutti, et al. 2016. "The Scenario Model Intercomparison Project (ScenarioMIP) for CMIP6." *Geoscientific Model Development* 9 (9): 3461–82.

Shiogama, Hideo, Shinichiro Fujimori, Tomoko Hasegawa, Michiya Hayashi, Yukiko Hirabayashi, Tomoo Ogura, Toshichika Iizumi, Kiyoshi Takahashi, and Toshihiko Takemura. 2023. "Important Distinctiveness of SSP3–7.0 for Use in Impact Assessments." *Nature Climate Change* 13 (12): 1276–78.

Swaminathan, Ranjini, Jacob Schewe, Jeremy Walton, Klaus Zimmermann, Colin Jones, Richard A. Betts, Chantelle Burton, et al. 2024. "Regional Impacts Poorly Constrained by

Climate Sensitivity." *Earth's Future* 12 (12). https://doi.org/10.1029/2024ef004901.

Watanabe, Masahiro, Sarah M. Kang, Matthew Collins, Yen-Ting Hwang, Shayne McGregor, and Malte F. Stuecker. 2024. "Possible Shift in Controls of the Tropical Pacific Surface Warming Pattern." *Nature* 630 (8016): 315–24.

Zelinka, Mark D., Timothy A. Myers, Daniel T. McCoy, Stephen Po-Chedley, Peter M. Caldwell, Paulo Ceppi, Stephen A. Klein, and Karl E. Taylor. 2020. "Causes of Higher Climate Sensitivity in CMIP6 Models." *Geophysical Research Letters* 47 (1). https://doi.org/10.1029/2019gl085782.

---

## Author Comment (AC3)

**ISIMIP3b paper review comments**

**Reviewer 1**

This paper describes the setup of the scenario and the climate-related forcings for ISIMIP3b. I truly appreciate the significant efforts made by the authors. The paper is important, interesting, and well written. I have a few comments.

**Major comment**

L511-525: The five representative ESMs include a hot model (UKESM1-0-LL). It has been suggested that the "hot model" issue of the CMIP6 ensemble can cause overestimation of impact assessments (Hausfather et al. 2022), Shiogama et al. 2022a). It depends on the variables and regions whether the hot model overestimates future change projections (Tokarska et al. 2020, Paik et al. 2023, McDonnell et al. 2024, Swaminathan et al. 2024, Li et al. 2024, Shiogama et al. 2022b, 2024, 2025). Although the internal variability component of the tropical Pacific surface warming pattern can affect the evaluation of "hot models" (Liang, Gillett, and Monahan 2024), the relative contributions of forced changes and internal variability to the observed tropical Pacific surface warming pattern are highly uncertain (Watanabe et al. 2024). Therefore, at least, please discuss the possible influence of the "hot model" issue on impact assessments of ISIMIP3b.

**Reply:**

Thanks so much for this very valid comment. To address the issue (and a related comment by reviewer 2), we added the following discussion to the relevant section:

"The five GCMs provide a good representation of both the mean and the range of the full CMIP6 multi-model ensemble ECS. According to (Meehl et al. 2020), the CMIP6 multi-model mean ECS is 3.7°C, which is precisely met by the mean ECS of the five ISIMIP3b GCMs. The transient climate response (TCR) of 2.0°C is also precisely met. This provides an improvement over ISIMIP2b, in the sense of the selected GCM subset reflecting the statistics of the larger CMIP ensemble. In ISIMIP2b the mean ECS for the full CMIP5 was 3.2°C compared with a mean ECS of 3.72°C for the four ISMIP2b GCMs (see Table S1 and S2 in (Jägermeyr et al. 2021)). The ISIMIP3b ensemble includes three models with below-average ECS (GFDL-ESM4, MPI-ESM1-2-HR, MRI-ESM2-0) and two models with above-average ECS (IPSL-CM6A-LR, UKESM1-0-LL) (see **Table 4**). In line with their ECS values, we find GFDL-ESM4 and UKESM1-0-LL to project the weakest and strongest global warming, respectively, under any future scenario considered (see **Figure 3**). Under SSP5-8.5, the global mean near-surface temperature in 2100 is about 3°C larger in UKESM1-0-LL than in GFDL-ESM4. Under SSP1-2.6, the projections are about 1.5°C apart. The ensemble mean warming of the ISIMIP3b CMIP6 models is significantly higher than the warming of the ISIMIP2b CMIP5 models, across global land area by an average of 0.3°C, but over the main breadbasket cropland regions by more than 0.5°C between 1983–2013 and 2069–2099, under both SSP1-2.6 and SSP5-8.5 (Table S1 in (Jägermeyr et al. 2021). This is in line with the higher median ECS in CMIP6 compared to CMIP5; indeed, some CMIP6 models have an ECS above the assessed likely (2.5°C to 4°C) and very likely (2°C to 5°C) ranges in the IPCC's sixth assessment report (AR6) (Forster et al. 2021). The reasons for these higher estimates of ECS are

complex, with cloud feedback processes playing an important role (Zelinka et al. 2020). While the plausibility of the very high ECS estimates has been questioned, recent studies indicate CMIP6 models with high ECS tend to simulate cloud properties better than low ECS models (Bock and Lauer 2024); also, unaccounted natural variability may have biased the IPCC's assessed ranges somewhat low (Watanabe et al. 2024; Liang, Gillett, and Monahan 2024)

The ISIMIP3b ensemble reflects the spread in ECS of the overall CMIP6 ensemble, with two models above the AR6 likely range and one of these (UKESM1-0-LL) above the very likely range. The strong warming response of these models should be kept in mind when conducting ISIMIP3b-based impacts studies. However, depending on the region and variable of interest, the high ECS does not necessarily have any bearing on the magnitude or realism of projected regional impacts, and any further selection of models should not be based solely on ECS but on the models' suitability for the impacts variables in question (Swaminathan et al. 2024)). In many applications, results can be harmonized by describing the simulated impacts in terms of global mean temperature changes instead of time for the different emission scenarios."

**Minor comments**

L98: ISIMIP3 -> ISIMIP3b?

**Reply:** indeed, thanks!

Table 2 (page 13, pre-industrial control, 2015soc, 1st priority): Please omit "ensi".

**Reply:** Done!

Figure 4: Can you add the plot of bias correction data of the historical simulation? It would be a good example to show hot bias-correction reduced the bias.

**Reply:** Thanks for the great suggestion, we revised the figure and added the bias-adjusted historical simulation data, showing the agreement with the observational data (see Figure below, brown line vs. thick black line). While doing so we also discovered that the uncorrected climate model data (blue and green lines) was not correctly displayed, and corrected it.

[Figure]

**Figure 4:** Global multi-year daily mean near-surface relative humidity for UKESM1-0-LL historical (1979-2014) and SSP5-8.5 (2065-2100), with uncorrected historical simulated data in blue, uncorrected future simulated data in green, historical bias-adjusted data in purple and brown, future bias-adjusted data in red and orange, and observational reference data in black. The bias is effectively reduced throughout all days of the year (brown line closely matching the black line) when ISIMIP3BASD v2.5 is applied in running-window mode in steps of one day (BA2). In contrast, a month-by-month application, which was the only option in ISIMIP3BASD v1.0, generates discontinuities at each turn of the month (BA1).

L808: "the Global Surge and Tide Model (GTSM) model" -> the Global Surge and Tide Model (GSTM)"?

**Reply:** We have adjusted the text. The correct name is Global Tide and Surge Model (GTSM).

Bibliography

Bock, Lisa, and Axel Lauer. 2024. "Cloud Properties and Their Projected Changes in CMIP Models with Low to High Climate Sensitivity." *Atmospheric Chemistry and Physics* 24 (3): 1587–1605.

Forster, P., T. Storelvmo, K. Armour, W. Collins, J-L Dufresne, D. Frame, D. Lunt, et al. 2021. *Chapter 7: The Earth's Energy Budget, Climate Feedbacks, and Climate Sensitivity. Climate Change 2021: The Physical Science Basis*.

Hausfather, Zeke, Kate Marvel, Gavin A. Schmidt, John W. Nielsen-Gammon, and Mark Zelinka. 2022. "Climate Simulations: Recognize the 'hot Model' Problem." Nature Publishing Group UK. May 4, 2022. https://doi.org/10.1038/d41586-022-01192-2.

Jägermeyr, Jonas, Christoph Müller, Alex C. Ruane, Joshua Elliott, Juraj Balkovic, Oscar Castillo, Babacar Faye, et al. 2021. "Climate Impacts on Global Agriculture Emerge Earlier in New Generation of Climate and Crop Models." *Nature Food* 2 (11): 873–85.

Liang, Yongxiao, Nathan P. Gillett, and Adam H. Monahan. 2024. "Accounting for Pacific Climate Variability Increases Projected Global Warming." *Nature Climate Change* 14 (6): 608–14.

Meehl, Gerald A., Catherine A. Senior, Veronika Eyring, Gregory Flato, Jean-Francois Lamarque, Ronald J. Stouffer, Karl E. Taylor, and Manuel Schlund. 2020. "Context for Interpreting Equilibrium Climate Sensitivity and Transient Climate Response from the CMIP6 Earth System Models." *Science Advances* 6 (26): eaba1981.

Swaminathan, Ranjini, Jacob Schewe, Jeremy Walton, Klaus Zimmermann, Colin Jones, Richard A. Betts, Chantelle Burton, et al. 2024. "Regional Impacts Poorly Constrained by Climate Sensitivity." *Earth's Future* 12 (12). https://doi.org/10.1029/2024ef004901.

Watanabe, Masahiro, Sarah M. Kang, Matthew Collins, Yen-Ting Hwang, Shayne McGregor, and Malte F. Stuecker. 2024. "Possible Shift in Controls of the Tropical Pacific Surface Warming Pattern." *Nature* 630 (8016): 315–24.

Zelinka, Mark D., Timothy A. Myers, Daniel T. McCoy, Stephen Po-Chedley, Peter M. Caldwell, Paulo Ceppi, Stephen A. Klein, and Karl E. Taylor. 2020. "Causes of Higher Climate Sensitivity in CMIP6 Models." *Geophysical Research Letters* 47 (1). https://doi.org/10.1029/2019gl085782.

---

## Author Comment (AC4)

Upfront reply:

"Dear Juan,

thanks so much for your careful review and the very relevant issues you raise.

Please find our point-by-point response below.

Best greetings, on behalf of the author team,

Katja Frieler"

—------------------------------------------------------------------------------------------------------------------

Dear Juan Antonio Añel,

Thanks for pointing to this. We are very sorry as part of the issues may be due to a miscommunication from our side. Please find our individual responses below.

Kind regards

Martin and Katja

**Comment 1:** First, the section states "The MIT data on tropical cyclone tracks with wind and precipitation fields data shall be used for non-commercial research or academic purposes only. Data can be made available by the ISIMIP team upon written consent by Kerry Emanuel (MIT, email: emanuel@mit.edu)".  Dr. Emanuel is a co-author of this submitted manuscript, therefore, it does not make any sense to point out to readers to contact somebody that is co-author to get access to the mentioned data. Therefore, you must publish the mentioned data in one of the repositories acceptable according to our policy, and reply to this comment with its link and permanent identifier. If some kind of law or regulation prevents you of sharing such data, then you need to provide us with evidence of it, and we will study the possibility of granting you an exception to it.

**Response:** The MIT data on tropical cyclone tracks is produced and owned by WindRiskTech, LLC. As a scientist you have to sign a license to get the data for free. It prohibits the redistribution of the data without express permission from WRT. That is why we cannot fully openly provide the data. However, it is freely available for scientific use. WindRiskTech's business model is to use revenues from sales of data set to commercial interests to fund the provision of free data to researchers. This can only work with researchers signing data licenses. Without such an arrangement, researchers would have to pay for the data. We have worked on an automatisation of the process, such that the data is available online but can only be downloaded after signing the associated license (see attached).

A similar process also applies to the CHAZ track data that is freely available online but can only be downloaded after signing an agreement that the data will only be used for noncommercial purposes.

The wind and precipitation fields generated for the MIT tracks are fully openly available on the ISIMIP repository with an associated DOI. We have changed the statement accordingly.

**Comment 2:** Second, you state "All other input data described are available for participating modelers with a respective account from the DKRZ server." This information is not enough. We can accept the DKRZ servers as hosting service for the repositories containing the mentioned data, but you must provide specific information (links and permanent identifiers) for all of them. I am aware that you communicated internally before that "The described data includes >1000 data sets that are comprised in quite a number of DOIs. Should we cite all these DOIs here again?". If with your question you were referring to these datasets, the answer to your question if you have used them to produce your manuscript, is "yes". A potential solution for this is that you deposit a document containing such information in a permanent repository and you link in the manuscript such repository (with its link and permanent identifier).

**Response:** We are very sorry for the miscommunication that is due to the fact that the statement was written for an earlier version of the paper. Meanwhile all climate-related forcing data that has been generated is publicly available on the ISIMIP repository and associated DOIs have been released. The 'Code and data availability' statement has been updated accordingly (response to comment 3). The DOIs have also been added in red to the last column of Table 1. Where we do not provide the data but only suggest an approach we would like to follow to generate them in the future (depending e.g. on funding), we have modified the entries to highlight that these cases are different from the other ones. This refers to the coastal water levels and the de-biased oceanic forcings. The Table now looks like this (all changes highlighted in red):

**Table 1: Climate-Related Forcing datasets for ISIMIP3b.**

| Forcing | Status | Source, description |
|---|---|---|
| Climate-related forcings ('picontrol', 'historical', 'ssp126', 'ssp370', 'ssp585') | | |
| Atmospheric forcings ('picontrol', 'historical', 'ssp585', 'ssp370', 'ssp126') | | |
| Gridded atmospheric climate forcing | mandatory | Bias-adjusted data (pre-industrial climate, historical climate, and future projections for the SSP1-2.6, SSP3-7.0, and SSP5-8.5 scenarios) generated by GFDL-ESM4, IPSL-CM6A-LR, MPI-ESM1-2-HR, MRI-ESM2-0, and UKESM1-0-LL within CMIP6 (Lange & Büchner, 2021), see section **2.1** |
| Local atmospheric climate forcing for lakes | mandatory | Atmospheric data extracted from the data sets above for 72 lakes that have been identified within the lake sector as |

| | | |
|---|---|---|
| | | locations (grid cell of the ISIMIP 0.5° grid, ISIMIP3 local lake sites) where models can be calibrated based on observed temperature profiles and hypsometry within ISIMIP3b (depth and area) (Lange and Büchner 2021) . |
| Tropical cyclone tracks with wind and precipitation fields | mandatory | SAvailable on request (see section 2.2), samples of synthetic tropical cyclone tracks derived from the five CMIP6 GCMs considered within ISIMIP generated by two different statistical downscaling approaches, see section 2.2.

 MIT approach (Emanuel et al., 2008, Emanuel et al., 2025):
 ● pre-industrial climate from IPSL-CM6A-LR, MPI-ESM1-2-HR and MRI-ESM2-0 (all 1850-2014), and from UKESM1-0-LL (1950-2100)
 ● historical climate from IPSL-CM6A-LR, MPI-ESM1-2-HR, UKESM1-0-LL and GFDL-ESM4 (all 1850-2014), and from MRI-ESM2-0 (1950-2014).
 ● Future climate: ssp126 (2061-2100), ssp370 (2015-2100) and ssp585 (2015-2100) from IPSL-CM6A-LR, MPI-ESM1-2-HR, MRI-ESM2-0, UKESM1-0-LL, and ssp585 (2061-2100) from GFDL-ESM4.

 Two different configurations (SD and CRH, see section 2.2) of the Columbia HAZard model (CHAZ, Lee et al., 2018, Lee et al., 2025):
 ● pre-industrial climate (1601-2100) from GFDL-ESM4, IPSL-CM6A-LR, MPI-ESM1-2-HR, MRI-ESM2-0, and UKESM1-0-LL.
 ● historical climate (1850-2014) from GFDL-ESM4, IPSL-CM6A-LR, MPI-ESM1-2-HR, MRI-ESM2-0, and UKESM1-0-LL
 ● future climate (2015-2100): ssp126, ssp370, ssp585 from GFDL-ESM4, IPSL-CM6A-LR, MPI-ESM1-2-HR, MRI-ESM2-0, and UKESM1-0-LL.

 For tracks generated by the MIT approach, we also provide wind and precipitation fields (Quesada-Chacón et al., 2025) |
| Lightning | mandatory | Flash Rate Monthly Climatology not changing with climate change (Cecil, 2006) |

| Oceanic forcings ('picontrol', 'historical', 'ssp585', 'ssp370', 'ssp126') | | |
|---|---|---|
| Oceanic climate forcing | mandatory | Uncorrected data (pre-industrial climate, historical climate, and future projections for the SSP1-2.6, SSP3-7.0, and SSP5-8.5 scenarios) generated by GFDL-ESM4, IPSL-CM6A-LR, MPI-ESM1-2-HR, and UKESM1-0-LL within CMIP6 (Büchner 2024), see section **2.4** |
| Coastal water levels | | |
| Coastal water levels | mandatory | In section **2.3** we describe a method to generate relative sea level projections that smoothly extend tide gauge observations into the future building on a Bayesian model (Perrette & Mengel, 2025). For ISIMIP3, we plan to extend the framework to all coastlines and directly use ISIMIP GCM output for the global thermosteric and local sterodynamic components, adjusting the gridded simulations to associated observations to ensure a consistent transition from the historical period. Ice sheet and glacier contributions are incorporated through spatial fingerprints, while unresolved vertical land motion processes are estimated from residuals at tide gauges and extrapolated where no observations are available. We are also developing an approach to extend the sea level projections to daily maximum water levels derived from the ISIMIP3 atmospheric forcings (daily mean Surface Air Pressure and daily mean Near-Surface Wind Speed). |
| Atmospheric composition or fluxes | | |
| Atmospheric $CO_2$ concentration | mandatory | (Büchner & Reyer, 2022) based on the following sources: 1850-2005: (Meinshausen et al., 2011); 2006-2014: Global annual CO2 from NOAA Global Monthly Mean $CO_2$ (Lan et al., 2023); 2015-2100: (Meinshausen et al., 2020) |
| Atmospheric $CH_4$ concentration | mandatory | (Büchner & Reyer, 2022) based on the following sources: 1850-2014:(Meinshausen et al., 2017); 2015-2100: (Meinshausen et al., 2020) |
| Climate-Related Forcings for the sensitivity experiment 'varlightning', using above forcing data except for: | | |

| Lightning data ('varlightning') | | |
|---|---|---|
| Varying lightning according to climate change | mandatory | Lightning data has been generated for the ssp126, ssp370, and ssp850 climate projections from UKESM1-0-LL (Kaplan et al., 2023) |
| Climate-Related Forcings for the 'de-biased' sensitivity experiment | | |
| Global oceanic forcings | | |
| Oceanic forcings based on de-biased atmospheric forcings | mandatory | In section **2.4.2** we describe an approach to de-bias the oceanic forcings based on the ocean biogeochemistry model NEMO-PISCES forced by a de-biased version of the IPSL-CM6A-LR-based atmospheric forcing as an option to fulfil the demand for de-biased ocean data we would like to follow. |
| Regional oceanic forcings | | |
| De-biased oceanic forcing based on observed oceanic data for individual variables and regions | mandatory | The regional models of the fisheries and marine ecosystem sector have applied regional bias-adjustments within their impact simulations that are described in section **2.4.3** and that make these simulations part of the 'de-biased' sensitivity experiment in the sector (see Table 2) while the default experiments are based on the raw oceanic forcings. |

**Comment 3:** Also, you state that "Data will be made publicly available... at the ISIMIP data repository at https://data.isimip.org/". First, we can not accept expressions of future compliance with our policy. The policy of the journal is very clear that compliance with code and data availability must be assured before submitting a manuscript to the journal. Therefore, you must publish all the datasets, as they should have been published before submission, and your manuscript desk rejected instead of accepted for Discussions because of such lack of compliance.

**Response:** All the generated climate-related forcings for ISIMIP3 are now publicly available in the ISIMIP repository (also see response above). Where we only describe a method to generate the data in the future this is also clearly highlighted.

**Comment 4:** A major problem is that you host your data in the ISIMIP data portal, managed by the PIK. You state that it complies with the policy of the journal, but it does not comply. I have checked the ISIMIP webpage and terms of use. First, we do not have any kind of guarantee that

the PIK servers comply with our requirements for long-term archival, usually requested in at least 15-20 years of secured funding to operate. The ISIMIP portal is simply a subdomain of pik-postdam.de, operating under the same IP address than the PIK webpage. Second, you state that data removal has to be approved by the ISIMIP steering committee, but my understanding is that this is basically the list of authors contributing to this manuscript, which actually means that the authors can decide to remove the data, and this is not acceptable according to the policy of the journal. Also, the "ISIMIP data team" is mentioned, but it is not clear at all who are the persons in such a committee or how it operates. From the structure published in the ISIMIP web page, it looks like if only one person is involved on it, identified as the "ISIMIP data manager".

**Response:** Thanks for this comment! We have clearly missed the opportunity to clarify the governance and the operational rules of the ISIMIP repository in the 'code and data availability' statement. We rewrote the section and attached a confirmation letter by the  the hosting institution, i.e. Potsdam Institute for Climate Impact Research (PIK) e.V. that ensures the long-term sustainability of the repository. As a member of the TIB DOI Consortium PIK commits itself to adhering to the [DataCite Consortium Agreement](). This includes commitments to data persistence (§4 a.) as well as maintaining and updating metadata (§4 c.) and explicitly forbids "withdrawing content without posting a notification". According to these rules the ISIMIP repository includes a [system to document and trace back ]()data issues and updates including information on how to retrieve the archived data. Users who have registered for the 'ISIMIP data' list will be immediately informed by email. The data described in the paper are all covered by DOIs (see response above) which also allows for tracking of potential previous versions.

**Comment 4:** In summary, there are many outstanding issues regarding your manuscript and its compliance with the policy of the journal, and the fact that it is under review and Discussions without having properly addressed all of them is irregular. Therefore, please, publish the necessary data in one of the repositories listed in our policy and reply to this comment as soon as possible with a modified 'Code and Data Availability' section for your manuscript, which must include the relevant information (link and handle or DOI) of the new repositories, and which you should include in a potentially reviewed manuscript.

**Response:** We are sorry that we created the wrong impression that only part of the data is openly accessible. We published all data we refer to with DOI and cite them when mentioning them in the paper, so full citations of all data sets are now included in the reference list. We rewrote the "Code and Data Availability" section as follows:

**Code and data availability.** All generated ISIMIP3 climate-related forcing data described in this paper is publicly available at the [ISIMIP data repository]() The repository is hosted by the Potsdam Institute for Climate Impact Research (PIK) e.V. which is part of the [TIB DOI Consortium]() ensuring persistent, FAIR-compliant data publication, by committing to adhering to the [DataCite Consortium Agreement](). This includes commitments to data persistence (§4 a.) as well as maintaining and updating metadata (§4 c.), which forbids "withdrawing content without posting a notification". In compliance with these rules a [system to document and trace back data issues ]()has been implemented in the repository to comply with this requirement. Additionally, should PIK be unable to continue hosting the ISIMIP repository, it will take responsibility for coordinating a timely

transfer of the full repository and its DOI infrastructure to an appropriate, trusted archive or institutional partner to ensure uninterrupted access and citation continuity. DOIs are used to refer to datasets in a persistent way. Whenever a dataset is replaced a copy is kept on tape, and a new DOI is issued, while the old DOI is kept online with information on how to retrieve the archived data. Whenever we need to replace datasets, we will create a new version of the DOI, marked by a version number at the end. This ensures that every DOI references exactly the datasets, which where public at the time of registration. Detailed information can be found in the ISIMIP terms of use at https://www.isimip.org/gettingstarted/terms-of-use/ (ISIMIP terms of use, 2023).

- I must note that if you do not fix this problem, we will have to reject your manuscript for publication in our journal.

Juan A. Añel

Geosci. Model Dev. Executive Editor

**List of DOI for Table 1**

- Gridded atmospheric climate forcing & Local atmospheric climate forcing for lakes
  - Stefan Lange, Matthias Büchner (2021): ISIMIP3b bias-adjusted atmospheric climate input data (v1.1). ISIMIP Repository. https://doi.org/10.48364/ISIMIP.842396.1
- Lightning (external)
  - Cecil, D. (2006). LIS/OTD 0.5 Degree High Resolution Monthly Climatology (HRMC) [Data set]. NASA Global Hydrometeorology Resource Center Distributed Active Archive Center. https://doi.org/10.5067/LIS/LIS-OTD/DATA303 Date Accessed: September 2020
- Oceanic climate forcing
  - Matthias Büchner (2024): ISIMIP3b ocean input data (v1.5). ISIMIP Repository. https://doi.org/10.48364/ISIMIP.575744.5
- Atmospheric $CO_2$ concentration & Atmospheric $CH_4$ concentration
  - Matthias Büchner, Christopher P.O. Reyer (2022): ISIMIP3b atmospheric composition input data (v1.1). ISIMIP Repository. https://doi.org/10.48364/ISIMIP.482153.1
- Varying lightning according to climate change (external)
  - Kaplan, J. O., Koch, A., & Lau, K. H.-K. (2023). Estimated future global lightning strokes (2010-2100) (v1.0.0) [Data set]. Zenodo. https://doi.org/10.5281/zenodo.7511843
- Tropical cyclones

- ○ Kerry Emanuel, Dánnell Quesada-Chacón, Lisa Novak, Christian Otto (2025): ISIMIP3b tropical cyclone tracks (MIT). ISIMIP Repository. https://doi.org/10.48364/ISIMIP.682793
- ○ Dánnell Quesada-Chacón, Lisa Novak, Linn Hamester, Christian Otto (2025): ISIMIP3b tropical cyclone wind and rain fields (MIT). ISIMIP Repository. https://doi.org/10.48364/ISIMIP.779038
- ○ Chia-Ying Lee, Suzana J. Camargo, Adam Sobel, Michael K. Tippett, Dánnell Quesada-Chacón, Matthias Büchner, Lisa Novak, Christian Otto (2025): ISIMIP3b tropical cyclone tracks (CHAZ). ISIMIP Repository. https://doi.org/10.48364/ISIMIP.808980

---

## Author Comment (AC5)

Potsdam Institute for Climate Impact Research e.V. P.O. Box 60 12 03 · 14412 Potsdam

**Statement on the Sustainable Stewardship of the ISIMIP Repository**

As the Directorate of the Potsdam Institute for Climate Impact Research (PIK), we affirm our full institutional commitment to the long-term, sustainable hosting and curation of the **Inter-Sectoral Impact Model Intercomparison Project (ISIMIP)** repository, available at <a href="https://data.isimip.org">https://data.isimip.org</a>.

PIK has been the official host and technical steward of the ISIMIP data infrastructure since its inception. In support of persistent, FAIR-compliant data publication, PIK joined the **TIB DOI Consortium** in 2020. Since then, ISIMIP datasets have been published with **DOIs assigned by the ISIMIP data team**, in accordance with the **DataCite Consortium Agreement**.

This agreement commits us to the following obligations:

- Data Persistence (§4a): All datasets published via the ISIMIP repository are guaranteed to remain accessible under their assigned DOI, or, in case of retracted or replaced data, upon request for retrieval from tape.
- Metadata Maintenance (§4c): PIK ensures the integrity and timely updating of metadata records, in line with DataCite's requirement not to withdraw content without public notification.
- Tombstone Pages: In the event that any dataset must be deprecated or removed, PIK commits to maintaining publicly available tombstone pages with full metadata, preserving the scholarly record and traceability.
- Continuity Planning: Should PIK be unable to continue hosting the ISIMIP repository, we will take responsibility for coordinating a timely transfer of the full repository and its DOI infrastructure to an appropriate, trusted archive or institutional partner to ensure uninterrupted access and citation continuity.

This institutional commitment underscores our recognition of ISIMIP's critical role in international climate impact research, and of our responsibility to maintain its accessibility, reliability, and scientific value over the long term.

Potsdam Institute for Climate Impact Research e.V.

Member of the Leibniz Association

Potsdam location: Telegrafenberg

Berlin location: EUREF-Campus

www.pik-potsdam.de

Board of Directors: Prof. Dr. Ottmar Edenhofer Prof. Dr. Johan Rockström Dr. Bettina Hörstrup

Association registry: Nummer VR 1038 P District court Potsdam

Bank account:
Mittelbrandenburgische
Sparkasse Potsdam (MBS)
IBAN: DE69 1605 0000 3502 2355 29
BIC: WELADED1PMB

11 July 2025

**Dr. Martin Park**Project Coordinator,
ISIMIP Manager

Research Department III Transformation Pathways

martin.park@pik-potsdam.de

Prof. Dr. Ottmar Edenhofer Scientific Director Dr. Bettina Hörstrup Administrative Director